# Targeting the transcription factor YY1 is synthetic lethal with loss of the histone demethylase KDM5C

Qian Zheng[1,5], Pengfei Li[2,5], Yulong Qiang[1,5], Jiachen Fan[1], Yuzhu Xing[1], Ying Zhang[1], Fan Yang [ID][1✉], Feng Li [ID][1,3✉] & Jie Xiong [ID][4✉]

## Abstract

**An understanding of the enzymatic and scaffolding functions of epigenetic modifiers is important for the development of epigenetic therapies for cancer. The H3K4me2/3 histone demethylase KDM5C has been shown to regulate transcription. The diverse roles of KDM5C are likely determined by its interacting partners, which are still largely unknown. In this study, we screen for KDM5C-binding proteins and show that YY1 interacts with KDM5C. A synergistic antitumor effect is exerted when both KDM5C and YY1 are depleted, and targeting YY1 appears to be a vulnerability in KDM5C-deficient cancer cells. Mechanistically, KDM5C promotes global YY1 chromatin recruitment, especially at promoters. Moreover, an intact KDM5C JmjC domain but not KDM5C histone demethylase activity is required for KDM5C-mediated YY1 chromatin binding. Transcriptional profiling reveals that dual inhibition of KDM5C and YY1 increases transcriptional repression of cell cycle- and apoptosis-related genes. In summary, our work demonstrates a synthetic lethal interaction between YY1 and KDM5C and suggests combination therapies for cancer treatments.**

**Keywords** KDM5C; YY1; Chromatin Recruitment; Promoter; Cancer Therapy
**Subject Categories** Cancer; Chromatin, Transcription & Genomics

## Introduction

Epigenetic regulators change phenotype acquisition or gene expression without altering the DNA sequence. By conferring a fitness advantage, mutational or non-mutational epigenetic reprogramming has been implicated in the evolution of tumors during malignant progression and metastasis (Hanahan, 2022). In addition, numerous studies have shown that aberrant epigenetic alterations contribute to cancer therapeutic resistance (Hogg et al, 2020; Miranda Furtado et al, 2019; Zafar et al, 2021). Therefore, targeting epigenetic regulators is an attractive strategy for cancer therapy.

The histone demethylase KDM5C is a member of the SMC homolog family. By specifically removing one or two methyl groups from Lys4 in histone H3, the dynamic balance in histone H3K4 methylation is maintained. KDM5C carries a composite catalytic region in which JmjN and JmjC domains are located. JmjC is critical for catalyzing the demethylation of H3K4me2/3, while the JmjN domain interacts with the JmjC domain for the complete catalytic function of KDM5C. H514 is a highly conserved amino acid that is located in the catalytic core of the JmjC domain, and its mutation eliminates KDM5C demethylase activity. KDM5C can suppress transcription by reducing the level of H3K4me2/3 via its demethylase activity; however, whether modulation of H3K4 methylation status is the only mechanism or the predominant mechanism by which KDM5C regulates gene expression is unclear.

KDM5C is frequently mutated or aberrantly expressed in various cancer types. Several studies have indicated that KDM5C is involved in regulating different biological processes in tumor cells through its histone demethylase activity and functions as a tumor suppressor. For example, the high-risk HPV E6 protein degrades KDM5C in cervical cancer cells, which results in super enhancers activation of the proto-oncogenes c-MET and EGFR by changing the H3K4me3/H3K4me1 ratio (Chen et al, 2018). In clear cell renal cell carcinoma cells, KDM5C inhibits the expression of genes related to glycogen metabolism by reducing H3K4me3 levels at the promoters of these genes, while inactivating KDM5C mutation confers cell resistance to ferroptosis (Zheng et al, 2021). However, the therapeutic strategy for KDM5C-deficient tumors is unknown. Intriguingly, KDM5C has also been shown to promote tumorigenesis, although the exact mechanism is unclear; in general, these observations suggest that KDM5C potentially regulates gene expression via a distinct mechanism.

To better understand the mechanisms underlying how KDM5C regulates gene expression, we screened and thus identified Yin Yang 1 (YY1) as a KDM5C-interacting partner. Targeting YY1 in KDM5C-deficient tumor cells effectively suppressed cell proliferation and tumorigenesis. Chromatin Immunoprecipitation followed by next-generation sequencing (ChIP-seq) analysis with multiple engineered cell lines revealed that KDM5C is essential for YY1

[1]Department of Medical Genetics, TaiKang Medical School (School of Basic Medical Sciences), Wuhan University, 430071 Wuhan, China. [2]Inner Mongolia Key Laboratory of Molecular Pathology, Inner Mongolia Medical University, 010059 Huhhot, Inner Mongolia, China. [3]Hubei Provincial Key Laboratory of Allergy and Immunology, Wuhan University, 430071 Wuhan, China. [4]Department of Immunology, TaiKang Medical School (School of Basic Medical Sciences), Wuhan University, 430071 Wuhan, China. [5]These authors contributed equally: Qian Zheng, Pengfei Li, Yulong Qiang. ✉E-mail: yangfan2022@whu.edu.cn; fli222@whu.edu.cn; jiexiong@whu.edu.cn

    

chromatin recruitment, which is partially dependent on an intact JmjC domain in KDM5C. Simultaneous loss of KDM5C and YY1 led to significantly enhanced transcriptional inhibition of a number of essential cell fate genes, elevated the apoptosis rate and led to cell cycle arrest. Our study suggests a novel scaffolding function of epigenetic regulators and provides a rationale for the development of new combination treatments.

# Results

## YY1 is identified as a partner of KDM5C

Previous studies have shown that KDM5C plays diverse roles in cancer progression. Indeed, a systemic The Cancer Genome Atlas (TCGA) clinical data-based analysis clearly revealed that high KDM5C expression was positively or negatively correlated with cancer patient survival, depending on the cancer type (Fig. 1A), reinforcing the idea that KDM5C may exhibit a distinct regulatory effect on gene expression, prompting us to screen the interacting partners of KDM5C. Tandem affinity purification (TAP) was subsequently performed. We generated a HEK293T cell line that stably expressed SBP- and Flag-tagged KDM5C, and the lysates of these cells were incubated with streptavidin-coated beads. The bead-bound proteins were subsequently eluted with buffer containing biotin. Next, the captured proteins were incubated with anti-Flag beads, and the proteins bound to these beads were eluted and analyzed by mass spectrometry (MS). The results of the MS analysis indicated that a number of proteins potentially interact with KDM5C, including YY1 (Fig. 1B), the overexpression of which has been observed in the vast majority of cancers (Gao et al, 2019; van Leenders et al, 2007). Given the pivotal role played by YY1 in the transcriptional regulation of numerous genes (Meliala et al, 2020), we focused on the interaction between KDM5C and YY1. To confirm their interaction, mutual coimmunoprecipitation (Co-IP) assays were performed in HEK293T cells, and the results clearly showed that KDM5C bound to YY1 (Fig. 1C). Co-IP with endogenous proteins was also performed and the results confirmed that KDM5C interacts with YY1 in mammalian cells (Fig. 1D). In addition, we also performed the semi-endogenous Co-IP with YY1 and KDM5C-Flag in 769-P renal cancer cell lines. As shown in Fig. 1E, KDM5C clearly bound to YY1. To delineate the interaction, a series of KDM5C truncation mutants were constructed and used for further analysis (Fig. 1F). We found that multiple regions, including the JmjN and JmjC domains, which are critical for H3K4me2/3 removal, were also involved in the interaction between KDM5C and YY1 (Fig. 1G).

## YY1 depletion significantly inhibits the proliferation and tumorigenicity of KDM5C-deficient tumor

Many studies have shown that YY1 functions as an oncogene because its overexpression is observed in multiple cancers (Li et al, 2022; Qiao et al, 2019; Wang et al, 2022). YY1 tethers cofactors and promotes or represses transcription, depending on its interacting partner, which includes c-myc and p53 (Shrivastava et al, 1993; Sui et al, 2004), prompting us to investigate the role of the YY1 KDM5C interaction in tumorigenesis. To achieve this goal, we first

selected human kidney proximal tubular epithelial cell line (HK-2), and several renal cancer cell lines (ACHN, 769-P and RCC4) with different KDM5C and YY1 protein expression levels, as described in our previous studies (Chen et al, 2018; Zheng et al, 2021) (Fig. 2A). Interestingly, knocking down YY1 in KDM5C-deficient RCC4 tumor cells significantly reduced the colony formation ability of the cells, which was not observed in the KDM5C-proficient tumor cell lines ACHN and 769-P (Figs. 2B and EV1B–D). Similar results were also obtained in a CCK8-based tumor cell proliferation assay (Figs. 2C and EV1E,F). Notably, tumor growth was markedly inhibited when both of KDM5C and YY1 were depleted, whereas the loss of either KDM5C or YY1 only negligibly influenced ACHN cell growth (Fig. 2D). These results strongly indicate that KDM5C might be important for YY1-related tumorigenesis, suggesting that YY1 may be a vulnerable target in KDM5C-deficient tumor cells.

## KDM5C co-localizes with YY1 and contributes to YY1 genome-wide chromatin recruitment

The stimulatory effect of YY1 on tumor cell growth was at least partially dependent on the presence of KDM5C. Given that YY1 always promotes tumorigenesis by increasing oncogene transcription, it is unlikely that YY1 tethers KDM5C because KDM5C has H3K4me2/3 demethylase activity, which is associated with repressed gene expression. A recent study revealed that KDM5B, which is a member of the KDM5 demethylase family, recruited a methyltransferase to suppress antitumor immunity, prompting us to explore whether KDM5C has a scaffolding function similar to that of KDM5B (Zhang et al, 2021). We generated a stable KDM5C-depleted ACHN cell lines (Fig. EV1B), and ChIP-seq assays were subsequently performed. As shown in Fig. 3A, the intensity of the YY1 binding peak was reduced upon loss of KDM5C, implying that KDM5C might be a general regulator of YY1 activity by mediating its chromatin recruitment. Next, we asked to what extent KDM5C co-localizes with YY1. To evaluate the genomic co-occupancy of KDM5C and YY1, we performed the cleavage under targets and tagmentation (CUT&Tag) assay in the ACHN renal cell line, and the annotation of the YY1 and KDM5C peaks revealed an obvious overlap ($n = 3645$) between YY1 and KDM5C binding events (Fig. 3B). Further analysis revealed a significant overlap, with 7957 genes being concurrently bound by both YY1 and KDM5C (Hypergeometric test, $P = 4.87e\text{-}17$). These overlapped genes accounted for 56.32% of the total genes bound by KDM5C and 66.71% of the total genes bound by YY1. Moreover, a heatmap and a metaplot analysis revealed that the enrichment of YY1 was reduced after KDM5C was depleted in the peaks ($n = 3645$) where YY1 and KDM5C were directly related (Fig. 3C). Reduced enrichment of YY1 was also observed in HK-2 cells with KDM5C depletion (Figs. EV1A and EV2A). These results revealed that the intensity of the YY1 binding peaks was partially dependent on the KDM5C protein level in both nontumoral cell and tumor cells, suggesting that KDM5C mediated YY1 chromatin recruitment may represent a universal mechanism of action. The localization of the KDM5C-dependent YY1 binding peaks was subsequently analyzed, and the results revealed that most of the peaks were enriched in promoters (Fig. 3D), implying that KDM5C may regulate gene expression via a distinct mechanism.

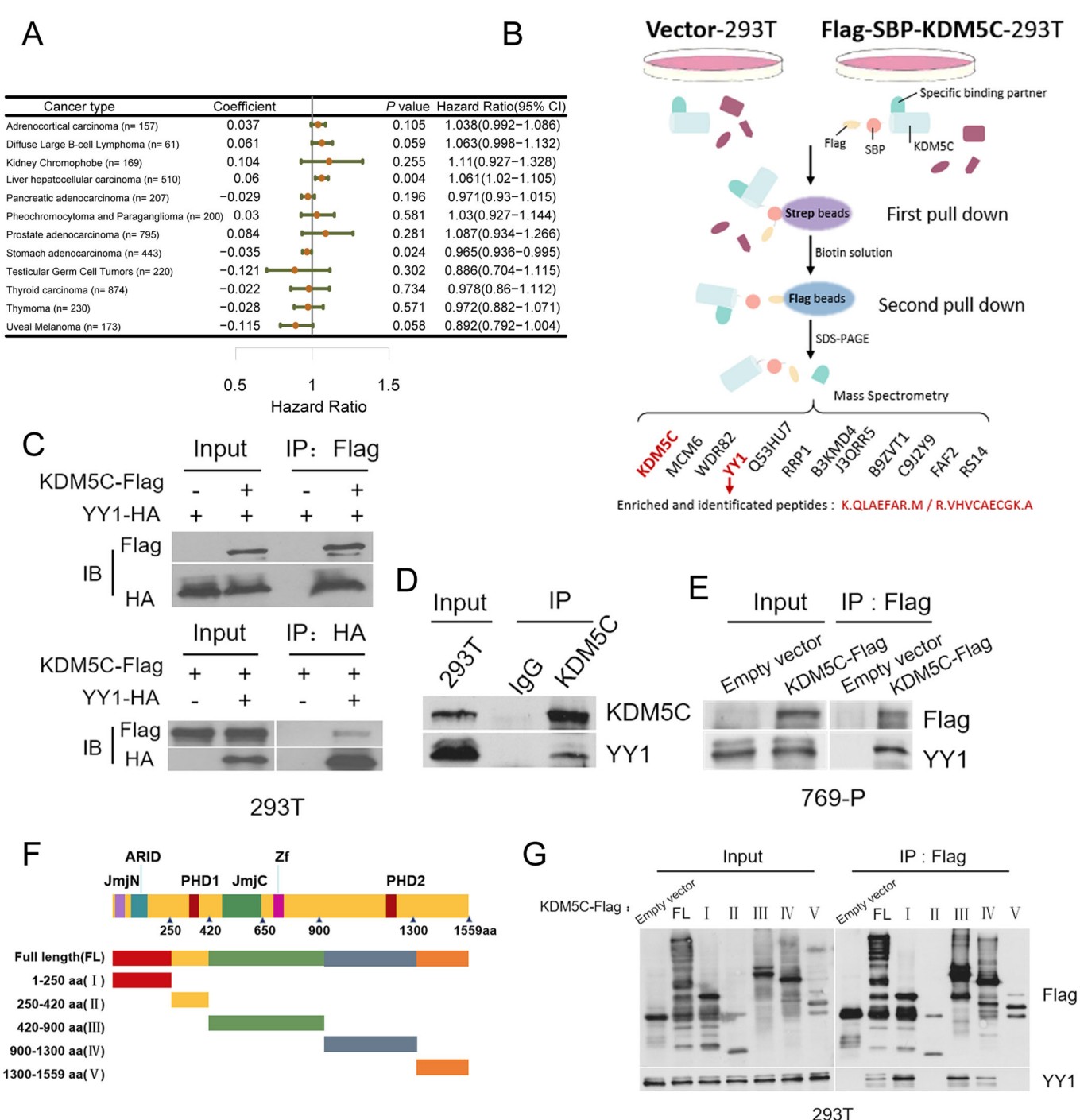

**Figure 1. YY1 was identified as a KDM5C binding protein.**

(A) Forest plot of the hazard ratio (HR) showing the association between high expression levels of KDM5C and overall survival in multiple cancers. For each type of cancer, the HR and 95% confidence interval (CI) are plotted as red dots and horizontal lines on the basis of a Cox proportional hazards model; when the HR = 1, the study object had no effect on overall survival; when the HR > 1, the study object was determined to be a risk factor; and when the HR < 1, the study object was identified as a protective factor. (B) Strategy for identifying KDM5C-binding partners in HEK293T cells via TAP-MS. (C, D) Co-IP followed by western blotting was used to detect the interaction between KDM5C and YY1 exogenously (C) and endogenously (D) in HEK 293 cells. (E) Semi-endogenous Co-IP with KDM5C-Flag and YY1 in the renal cancer cell line 769-P. (F) Schematic representation of KDM5C domains and corresponding constructs. (G) Semi-endogenous Co-IP with KDM5C truncation mutants and endogenous YY1 in HEK293T cells. Source data are available online for this figure.

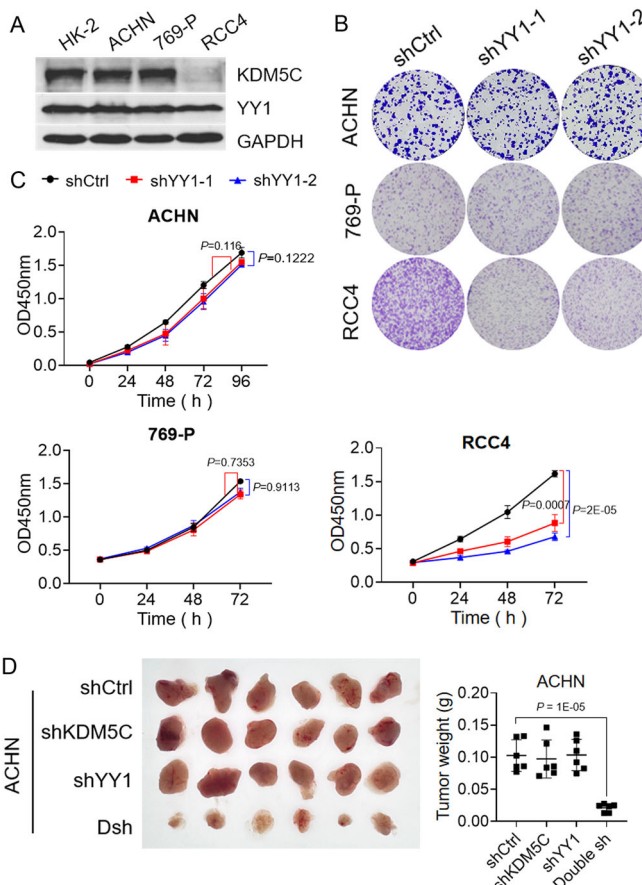

**Figure 2. YY1 is a vulnerable target in KDM5C-deficient tumor cells.**

(A) Western blotting showing KDM5C expression in HK-2 human kidney proximal tubular epithelial cells and renal cancer cell lines (ACHN, 769-P and RCC4). (B) Effect of YY1 knockdown on the survival of the indicated tumor cell lines as measured by a colony formation assay (n = 3 biological replicates). (C) Proliferation of the indicated cell lines after YY1 was depleted, as measured by a CCK8 assay. The data are shown as the mean ± SD of three independent experiments after analysis by one-way ANOVA. In ACHN cells, P = 0.116 (shYY1-1) and P = 0.1222 (shYY1-2); in 769-P cells, P = 0.7353 (shYY1-1) and P = 0.9113 (shYY1-2); and in RCC4 cells, P = 0.0007 (shYY1-1) and P = 2E-05 (shYY1-2). (D) Xenograft tumors excised from nude mice (n = 6/group) that had been subcutaneously injected with the indicated tumor cells. Statistical significance was determined by one-way ANOVA. P = 1E-05. Source data are available online for this figure.

## An intact catalytic domain but not the demethylation activity of KDM5C is required for YY1 recruitment

Since our results suggested that KDM5C functions as a scaffold and that KDM5C is essential for YY1 binding to a promoter, we asked, whether the demethylase activity of KDM5C is involved in YY1 recruitment. We generated ACHN cell lines that stably expressed KDM5C-WT or its enzymatically inactive mutant H514 A and performed ChIP-seq to evaluate the effects of KDM5C or its mutants on YY1 recruitment (Fig. EV1B). As shown in Fig. 4A, ectopic KDM5C WT but not the KDM5C-H514A mutant enhanced YY1 chromatin binding. A similar trend was also observed in HK-2 cells (Fig. EV2A). Because the primary function of KDM5C is to catalyze the removal of methyl groups from H3K4me2/3, we

evaluated whether H3K4me3 abundance plays a role in YY1 recruitment.

We first performed ChIP-seq of H3K4me3 in ACHN shKDM5C cells and analyzed the change in the signal intensity of this mark. Increased H3K4me3 enrichment was clearly observed in the KDM5C/YY1 overlapping region (Fig. 4B), but the opposite trend of YY1 recruitment was observed in KDM5C-depleted cells (Fig. 3C). To further understand the relationship between YY1 recruitment and H3K4me3, we focused on the overlapping regions of YY1 and H3K4me3 (n = 11,087), which were classified into KDM5C demethylase-dependent peaks (n = 9943) and KDM5C demethylase-independent peaks (n = 1144), as shown in Fig. 4C. Reduced YY1 ChIP-seq signals were observed in almost all of the overlapping YY1 and H3K4me3 regions (Fig. 4D,E), regardless of the effect of KDM5C on H3K4me3 intensity. We therefore hypothesized that an intact KDM5C JmjC domain, rather than KDM5C histone demethylase activity, is essential for YY1 recruitment.

To further explore the mechanism underlying the effects of KDM5C, we evaluated the interaction between KDM5C-H514A and YY1 and found that the mutation did not influence their interaction (Fig. EV3), indicating that the effect of KDM5C-H514A on YY1 chromatin recruitment was unlikely to be caused by disrupted protein binding activity. Because the catalytic composite JmjN/C domain binds the substrate H3 tail harboring methylated H3K4 and since a previous study revealed that none of the KDM5C PHD domains exhibited strong binding affinity for unmethylated or methylated histones (Iwase et al, 2007), we speculated that the replacement of the histidine residue at position 514 with an alanine residue destroyed the integrity of the JmjN/C domain and influenced the chromatin-binding activity of KDM5C. To test this hypothesis, we expressed KDM5C-WT or KDM5C-H514A in HEK293T cells and isolated chromatin-bound proteins. Interestingly, although the protein levels in whole-cell lysates were indistinguishable, H154A exhibited obviously reduced chromatin-binding capacity (Fig. 4F), which might help explain why KDM5C-H514A failed to recruit YY1 to chromatin. In addition, tumor cell proliferation was markedly inhibited only when both of KDM5C and YY1 were depleted in the KDM5C-proficient tumor cells ACHN and 769-P (Figs. 4G,H and EV1B,C). The inhibitory effect was blunted when KDM5C-WT was re-expressed; in contrast, KDM5C-H514A only moderately restored the growth of ACHN and 769-P double knockdown cells. A similar trend was also observed in the colony formation assay (Fig. 4J). In KDM5C-deficient RCC4, cells overexpression of KDM5C-WT largely restored cell sensitivity to YY1 depletion (Figs. 4I and EV1D). These results collectively suggest that an intact catalytic domain of KDM5C is required for YY1 recruitment.

## KDM5C cooperates with YY1 to regulate essential cell fate gene expression

To systematically evaluate the effect of KDM5C-mediated YY1 recruitment on gene expression, we performed RNA-seq with shKDM5C, shYY1 or double-knockdown ACHN cells. As shown in Fig. 5A,B, the loss of KDM5C or YY1 led to the upregulated or downregulated expression of several genes. In contrast, the simultaneous loss of KDM5C and YY1 resulted in altered transcription of thousands of genes (1511 genes whose expression

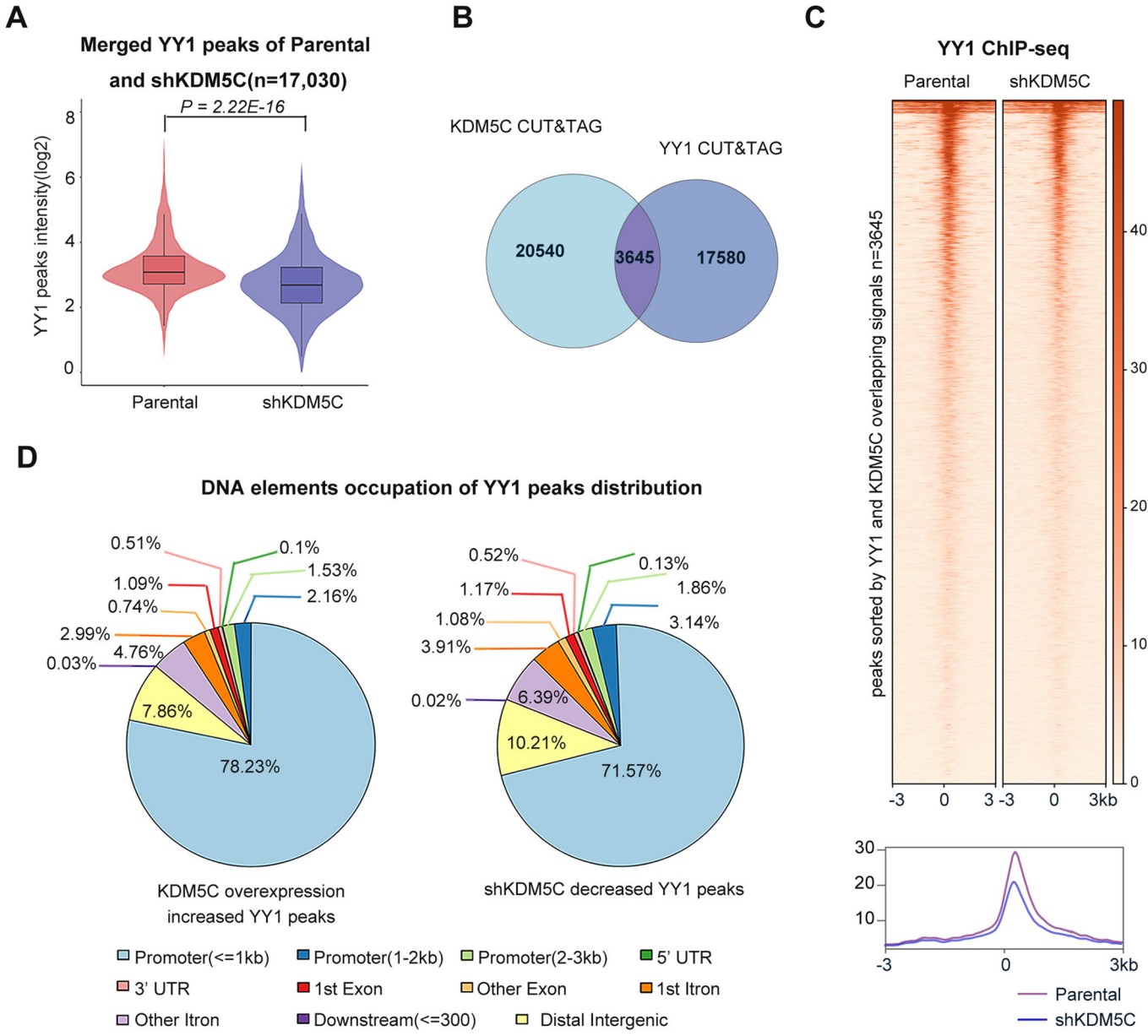

**Figure 3. KDM5C is essential for YY1 genome-wide chromatin recruitment.**

(A) Violin plot showing the alteration in YY1 binding peaks after KDM5C depletion. YY1 binding peaks in both ACHN parental and shKDM5C cells were counted ($n = 17,030$). The data are presented as mean ± SD. Statistical significance was determined by the Mann–Whitney $U$ test, $P = 2.22E\text{-}16$. (B) Venn diagram showing the overlapping peaks of YY1 and KDM5C in the co-occupied regions. Representative data from two independent CUT-Tag experiments. (C) Heatmap and metaplot showing the chromatin enrichment of YY1 at overlapping YY1 and KDM5C peak regions ($n = 3645$) in ACHN parental and shKDM5C cells. (D) Pie chart showing the annotated genomic distribution of altered YY1 peaks on different DNA elements in shKDM5C or KDM5C overexpressing cells. Source data are available online for this figure.

was downregulated versus 797 genes whose expression was upregulated, $|\log_2 FC| \geq 1.5$, $P$ value < 0.05, Fig. 5C), indicating that YY1 recruitment by KDM5C likely contributed to gene expression activation. A similar pattern was also observed via RNA-seq with HK-2 nontumoral cells (Fig. EV4). Compared with the loss of KDM5C or YY1 alone, double knockdown caused transcriptional repression of a sharply increased number of genes (1099 newly identified downregulated genes in total), as shown in Fig. 5D. Given that depletion of both YY1 and KDM5C led to significant suppression of tumor cell proliferation (Fig. 4), we next investigated

the gene expression patterns by analyzing the RNA-seq results and found that the expression of several cell cycle- and cell apoptosis-related genes, including JAM3, KNTC1, SMC2, SMC4, LMNB1 and FOXM1, was markedly downregulated in the double-knockdown cell group (Fig. 5E). These results were verified by qPCR in ACHN cells (Fig. 5F).

We compared the RNA-seq results with our CUT&Tag data, and found that almost half of the downregulated genes (49.96%) in the KDM5C/YY1 double-knockdown group were annotated in the genomic regions where YY1 and KDM5C overlapped,

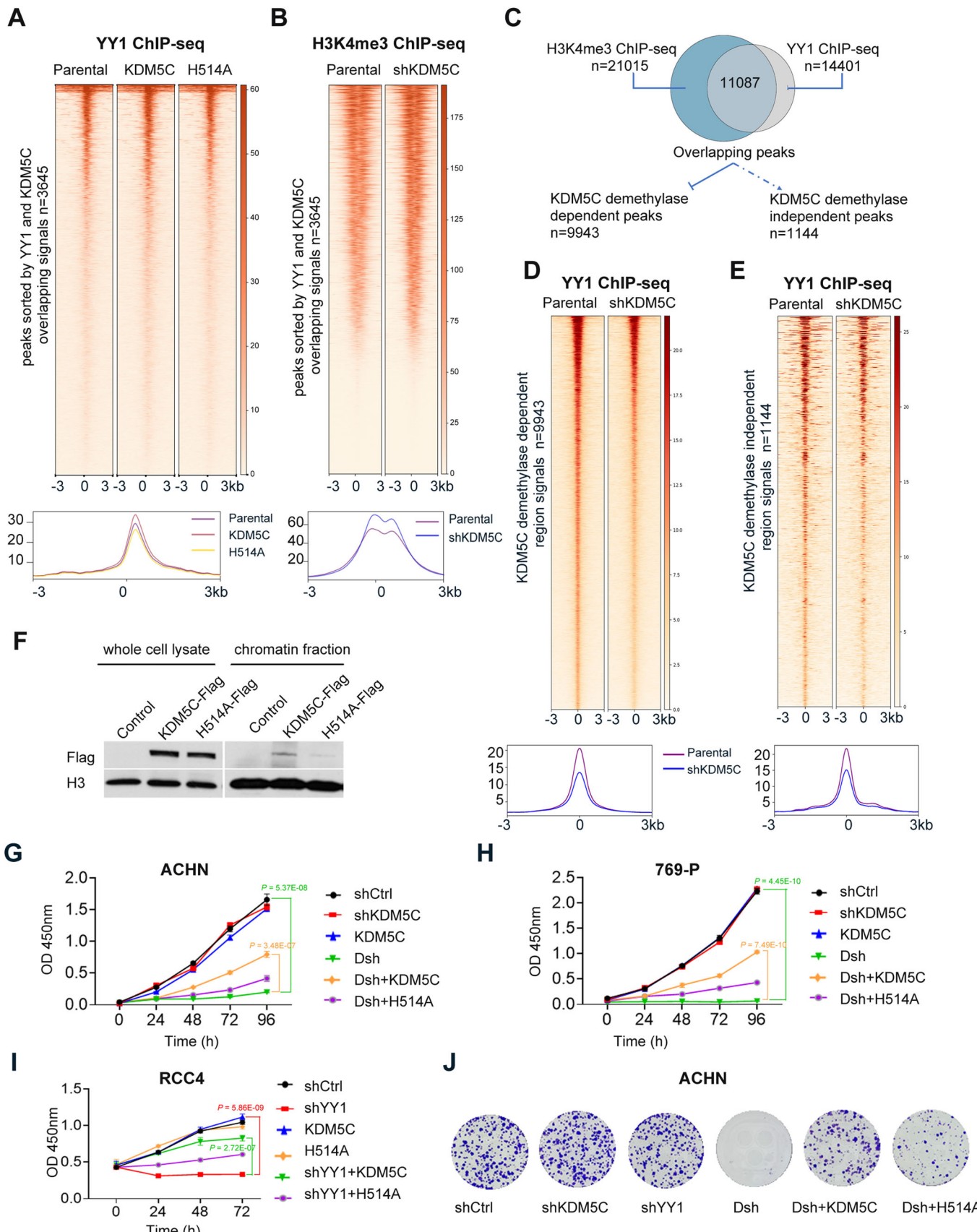

Figure 4.   An intact JmjC domain but not KDM5C H3K4 demethylation activity is required for KDM5C-mediated YY1 recruitment.

(A) Heatmap and metaplot showing the distribution of YY1 peaks in the overlapping regions of YY1 and KDM5C ($n = 3645$) in ACHN cells expressing the KDM5C-WT or KDM5C-H514A mutant. (B) Loss of KDMC5 resulted in increased H3K4me3 signal intensity in the YY1 and KDM5C overlapping region ($n = 3645$) in ACHN parental and shKDM5C cells. (C) Venn diagram showing the number of YY1 peaks in the H3K4me3/YY1 overlapping region ($n = 11,087$) with or without modulation by KDM5C. (D, E) Heatmaps and metaplots revealed reduced YY1 enrichment at KDM5C demethylase-dependent regions ($n = 9943$) or KDM5C demethylase-independent regions ($n = 1144$) in ACHN parental and shKDM5C cells. (F) KDM5C protein levels in whole-cell lysates or chromatin fractions from HEK293T cells transfected with KDM5C-WT or KDM5C-H514A mutant construct. (G–I) Effect of the re-expression of the KDM5C-WT or KDM5C-H514A mutant on the proliferation of ACHN (G), 769-P (H) or RCC4 (I) cells after the depletion KDM5C, YY1 or both, as determined by the CCK8 assay. Each experiment was repeated three times with similar results. "Dsh" indicates cells in which KDM5C and YY1 were both knocked down. The data are shown as the mean ± SD of three independent experiments after analysis via Student's $t$ test. In ACHN cells, $P = 5.37E-08$ (Dsh) and $P = 3.48E-07$ (Dsh+KDM5C); in 769-P cells, $P = 4.45E-10$ (Dsh) and $P = 7.49E-10$ (Dsh+KDM5C); in ACHN cells, $P = 5.86E-09$ (Dsh) and $P = 2.72E-07$ (Dsh+KDM5C). (J) Colony formation assay with the indicated cell lines ($n = 3$). "Dsh" indicates cells with KDM5C and YY1 simultaneously knocked down. Source data are available online for this figure.

as shown in Fig. EV5B. We then analyzed the YY1 peaks at the promoters of individual genes via the IGV browser. As shown in Fig. EV5C,D, KDM5C depletion reduced theYY1 peak intensity at the promoters of the indicated genes, and the presence of KDM5C increased the amount of YY1 recruited to these promoters. KDM5C-H514A marginally increased the YY1 intensity in the indicated regions. Moreover, ChIP-qPCR was performed in HK-2 cells and the result revealed that YY1 enrichment at the promoter region of the indicated genes was increased after KDM5C-WT was expressed, whereas KDM5C-H514A failed to increase YY1 binding (Fig. EV2C). These results strongly suggest that KDM5C increases the expression of essential cell-fate genes by recruiting YY1 to their promoters.

## Simultaneous loss of KDM5C and YY1 significantly promotes cell cycle arrest and cell apoptosis

Previous studies have shown that *SMC2/4, LMNB1*, and *KNTC1* are involved in cell cycle arrest and apoptosis regulation. To confirm the synergistic effect of KDM5C and YY1 on tumorigenicity and cancer cell proliferation, we analyzed the cell cycle progression and apoptosis rate of tumor cells in which KDM5C, YY1 or both had been depleted. We first measured the ratio of renal cancer cells in each phase of the cell cycle, and the results indicated that the loss of both KDM5C and YY1 caused an increased ratio of 769-P tumor cells arrested in the G2/M phase. Similar results were also observed in KDM5C-deficient RCC4 cells after YY1 depletion (Fig. 6A). Moreover, double knock-down of KDM5C and YY1 but not loss of either gene individually greatly increased the percentage of apoptotic cells in 769-P tumors. In addition, the absence of YY1 in RCC4 cells induced a significant increase in the percentage of apoptosis cells (Fig. 6B). These results confirmed the inhibitory effect of YY1 depletion on KDM5C-deficient cells (Fig. 4). Taken together, our study revealed a synergistic regulatory effect of KDM5C and YY1 on cancer cell proliferation and cell cycle progression.

## Discussion

As a member of the GLI-Kruppel family of zinc finger DNA-binding proteins, YY1 has been proven to regulate genes either by directly binding to a promoter or indirectly by associating with chromatin-remodeling proteins or histone modifiers (Gordon et al, 2006; Khachigian, 2018). YY1 is frequently overexpressed in many cancer types and reshapes the oncogenic gene expression network (Bonavida and Kaufhold, 2015; Sarvagalla et al, 2019). Although

YY1 can directly bind to a promoter DNA sequence, the mechanism underlying its recruitment to chromatin is still not fully understood. In addition, although targeting YY1 might be an attractive therapeutic strategy, a robust biomarker to analyze its effect is lacking. In this study, we found that YY1 is a KDM5C-associated protein, and the JmjC domain of KDM5C was essential for the KDM5C–YY1 interaction. Moreover, we demonstrated that KDM5C mediates the chromatin recruitment of YY1, which is important for YY1 binding to promoter regions and the transcription of cell fate-related genes. Importantly, targeting YY1 effectively inhibited the proliferation and tumorigenicity of tumor cells with low KDM5C expression. Considering these findings, we propose the following model: YY1 chromatin recruitment is at least partially dependent on the presence of KDM5C, and in tumor cells with high KDM5C expression, a small amount of YY1 can be recruited to chromatin even after KDM5C is knocked down. In contrast, depleting YY1 in low KDM5C-expressing tumor cells completely eliminated chromatin-bound YY1 and promoted tumor regression (Fig. 7).

As a specific H3K4 histone demethylase, KDM5C plays dual roles as both a proto-oncogene and tumor suppressor (Chen et al, 2018; Ji et al, 2015; Shen et al, 2016; Shen et al, 2021). KDM5C interacts with specific transcription factors, such as MYC (Outchkourov et al, 2013; Secombe et al, 2007), and downregulates gene expression by reducing H3K4me2/3 levels at enhancer or promoter regions. However, studies, including our previous reports, have shown that KDM5C also upregulates the expression of certain genes, but the relevant molecular mechanism of KDM5C effect on gene expression is unclear (Outchkourov et al, 2013; Scandaglia et al, 2017; Zheng et al, 2021). A recent study showed that the histone demethylase KDM5B recruited the H3K9 methyltransferase SETDB1 to suppress antitumor immunity in a KDM5B demethylase-independent manner, implying that the scaffolding function of this epigenetic regulator was important in certain contexts, a finding that has been appreciated to a lesser degree in prior studies (Zhang et al, 2021). Our finding that KDM5C recruited YY1 to chromatin in a histone demethylase-independent manner reinforces the evidence suggesting that the scaffolding function of epigenetic regulators might be critical for tumorigenesis. Indeed, epigenetic inhibitors targeting the catalytic domain often show limited efficacy and intolerable toxicity, and our findings strongly suggest that targeting the scaffolding function should be considered in the development of new cancer prevention strategies and therapies.

Abnormalities in enzymes that regulate epigenetic modifications are important drivers of tumor occurrence and cancer progression (Dawson, 2017). Therefore, epigenome-based therapies might be

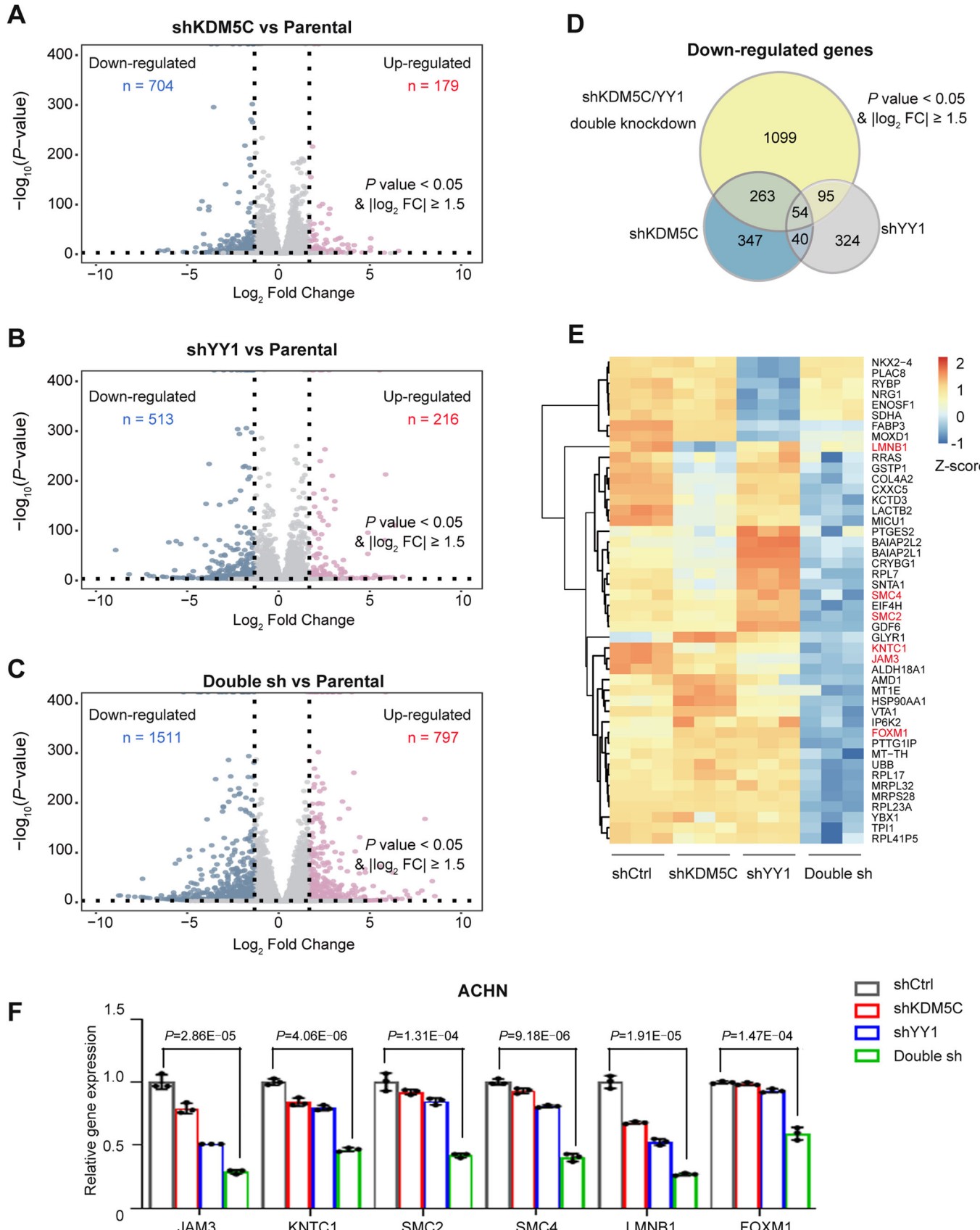

**Figure 5. KDM5C and YY1 coregulate the expression of a number of essential cell-fate genes.**

(A–C) Volcano plots showing the change in the number of transcripts in shKDM5C, shYY1 or double shKDM5C ACHN cells, as determined by RNA-seq analysis ($|\log_2$ FC $| > 1.5$, $P$ value $< 0.05$). "Double sh" indicates cells with KDM5C and YY1 simultaneously knocked down. (D) Venn diagrams showing the number of downregulated genes in shKDM5C, shYY1 or shKDM5C/YY1 double-knockdown cells. (E) Heatmap showing the normalized gene expression profile in shKDM5C, shYY1 or double sh ACHN cells. "Double sh" indicates cells with KDM5C and YY1 simultaneously knocked down. (F) qPCR was used to assess the transcription of select genes in the indicated ACHN cells. Each experiment was repeated three times with similar results. "Double sh" indicates cells with KDM5C and YY1 double knocked down. The data are shown as the mean ± SD of three independent experiments after analysis by two-tailed Student's $T$ test. $P = 2.86E-05$ (JAM3), $P = 4.06E-06$ (KNTC1), $P = 1.31E-04$(SMC2), $P = 9.18E-06$(SMC4), $P = 1.91E-05$(LMNB1) and $P = 1.47E-04$ (FOXM1). Source data are available online for this figure.

developed to block the dysregulated expression of proto-oncogenes in tumor cells and increase antitumor responses (Miranda Furtado et al, 2019; Nebbioso et al, 2018). KDM5C has been shown to be frequently mutated or aberrantly expressed in various cancers. The degree of YY1 chromatin recruitment was decreased in KDM5C-deficient tumor cells, and further inhibition of YY1 expression might completely suppress YY1-directed transcription and significantly impair the proliferation of tumor cells. Furthermore, simultaneously targeting KDM5C and YY1 effectively inhibited tumor growth in KDM5C-proficient tumor cells. In recent years, the induction of synthetic lethality has been applied to the identification of novel drug targets, opening new avenues to tumor-targeted therapy (Nijman and Friend, 2013). Considering the definition of synthetic lethality and the results presented here, we propose that KDM5C and YY1 form a synthetically lethal gene pair, providing a rationale for the clinical treatment of patients with KDM5C-deficient cancer by targeting YY1.

In summary, our work provides mechanistic insight into YY1 chromatin recruitment and suggests an important role for the scaffolding function of the histone demethylase KDM5C. Because proteolysis-targeting chimeras (PROTACs) are used to target and degrade proteins with important scaffolding functions and serve as a potential therapeutic strategy in cancer, future work focusing on the design and synthesis of KDM5C PROTAC drugs is important, and further studies in this research direction are warranted.

# Methods

### Reagents and tools table

| Reagent/resource | Reference or source | Identifier or catalog number |
| --- | --- | --- |
| **Experimental models** | | |
| Human kidney proximal tubular epithelial cell line HK-2 | American Type Culture Collection | |
| Human renal cancer cell lines (ACHN, 769-P and RCC4) | American Type Culture Collection | |
| HEK293T | Institute of Cellular Resources, Shanghai Institutes for Biological Sciences, Chinese Academy of Sciences. | |
| BALB/c nude mice | Beijing Vital River Laboratory Animal Technology (China) | |
| **Recombinant DNA** | | |
| pHAGE-HA-Flag-Ctrl | Our Lab | |
| pHAGE-HA-KDM5C | Our Lab | |
| pHAGE-Flag-KDM5C | Our Lab | |
| pHAGE-HA-YY1 | Our Lab | |

| Reagent/resource | Reference or source | Identifier or catalog number |
| --- | --- | --- |
| pHAGE-Flag-YY1 | Our Lab | |
| pHAGE-HA-H514A | Our Lab | |
| pHAGE-Flag-H514A | Our Lab | |
| pHAGE-Flag-KDM5C-I | Our Lab | |
| pHAGE-Flag-KDM5C-II | Our Lab | |
| pHAGE-Flag-KDM5C-III | Our Lab | |
| pHAGE-Flag-KDM5C-IV | Our Lab | |
| pHAGE-Flag-KDM5C-V | Our Lab | |
| pHAGE-Flag-SBP-KDM5C | Our Lab | |
| pLKO.1-shCtrl | Our Lab | |
| pLKO.1-shCtrl-EGFP | Our Lab | |
| pLKO.1-shKDM5C-1 | Our Lab | |
| pLKO.1-shKDM5C-2 | Our Lab | |
| pLKO.1-shYY1-1 | Our Lab | |
| pLKO.1-shYY1-2 | Our Lab | |
| pLKO.1-EGFP-shKDM5C- 1 | Our Lab | |
| pLKO.1-EGFP-shKDM5C-2 | Our Lab | |
| pHAGE-Flag-KDM5C-Hygro | Our Lab | |
| pHAGE-Flag-H514A-Hygro | Our Lab | |
| **Antibodies** | | |
| Anti-KDM5C | Cell Signaling Technology | D29B9 |
| Anti-KDM5C | BETHYL | A301-034A |
| Anti-KDM5C | Proteintech | 14426-1-AP |
| Anti-YY1 | Cell Signaling Technology | D5D9Z |
| Anti-Flag | Proteintech | 20543-1-AP |
| Anti-HA | Proteintech | 66006-2-Ig |
| Anti-GAPDH | Proteintech | 60004-1-Ig |
| Anti-β-actin | Proteintech | 66009-1-Ig |
| Anti-H3 | Cell Signaling Technology | 2650 |
| Anti-H3K4me3 | Cell Signaling Technology | 9751 |
| Rabbit IgG | ABclonal | AC042 |
| **Oligonucleotides and other sequence-based reagents** | | |
| Plasmid construction primer sequences | This study | "Methods" section |
| qRT-PCR primer sequences | This study | "Methods" section |
| ChIP-qPCR primer sequences | This study | "Methods" section |
| **Chemicals, enzymes and other reagents** | | |
| Fetal bovine serum | Cell-box | AUS-01S-02 |

| Reagent/resource | Reference or source | Identifier or catalog number |
|---|---|---|
| NEOFECT DNA transfection reagent | NeoFect | TF20121201 |
| Polybrene | Solarbio | H8761 |
| Puromycin | Solarbio | P8230 |
| RIPA buffer | Beyotime | P0013B |
| Protease inhibitor mixture cocktail | MedChemExpress | HY-K0010 |
| Anti-HA magnetic beads | MedChemExpress | HY-K0201 |
| Anti-Flag magnetic beads | MedChemExpress | HY-K0207 |
| Hygromycin | Beyotime | ST-1389 |
| Matrigel Basement Membrane Matrix | Corning | 356234 |
| **Software** | | |
| GraphPad Prism software V9.4.1 | | |
| Flowjo | | |
| **Other** | | |
| MycoBlue Mycoplasma Detector | Vazyme | D101-01 |
| RNA isolater Total RNA Extraction Reagent | Vazyme | R401-01 |
| ReverTra Ace qPCR RT Master Mix with gDNA Remover | TOYOBO | FSQ-301 |
| ChamQ SYBR qPCR Master Mix | Vazyme | Q311-02 |
| Hyperactive Universal CUT&Tag Assay Kit for Illumina Pro | Vazyme | TD904 |
| TruePrep Index Kit V2 for Illumina | Vazyme | TD202 |
| Cell counting kit 8 | MedChemExpress | HY-K0301 |
| BCA protein assay kit | Beyotime | P0012 |
| Enhanced chemiluminescence detection kit | Vazyme | E411-05 |
| Annexin V-FITC/PI apoptosis kit | Multi Sciences | AP101 |
| Cell Cycle and Apoptosis Analysis Kit | Beyotime | C1025 |
| Flow cytometer | BD Bioscience | LSRFortessaX20 |
| Microplate reader | Epoch | BioTek |
| CFX Connect Real-Time PCR Detection System | Bio-Rad | |

## Vectors and plasmids

The pHAGE vector carries a puromycin resistance cassette, endowing the transfected cells with resistance to puromycin, hence enabling us to select and generate stable cells. The overexpression vectors used in our context (KDM5C or YY1 with HA or Flag tags and their truncated constructs) were constructed using pHAGE for mammalian cell expression. The KDM5C enzymatically inactive mutant H514A was generated via site-directed mutagenesis PCR from the pHAGE-KDM5C plasmid. shRNAs targeting human KDM5C and YY1 were constructed using pLKO.1. The construction of pLKO.1 is detailed on the *addgene* website. All of the constructs were validated by DNA sequencing. The targeted sequences of KDM5C and YY1 genes are as follows:

shCtrl: 5'-CAACAAGATGAAGAGCACCAA-3';
shKDM5C-1: 5'-AGTACCTGCGGTATCGGTATA-3';
shKDM5C-2: 5'-GCCACACTTGAG GCCATAATC-3';
shYY1-1: 5'-GACGACGACTACATTGAACAA-3';
shYY1-2: 5'-GCCTCTCCTTTGTATATTATT-3'.

## Cell culture

The human kidney proximal tubular epithelial cell line HK-2 and human renal cancer cell lines (ACHN, 769-P and RCC4) were obtained from the American Type Culture Collection (ATCC). HEK293T cells were obtained from the Institute of Cellular Resources, Shanghai Institutes for Biological Sciences, Chinese Academy of Sciences. HK-2, ACHN, RCC4, and HEK293T cells were cultured in high-glucose Dulbecco's modified Eagle's medium (DMEM, HyClone) while 769-P cells were cultured in Roswell Park Memorial Institute-RPMI medium (1640, HyClone) and both media were supplemented with 10% fetal bovine serum (FBS, Cell-box, AUS-01S-02) in a humidified atmosphere containing 5% $CO_2$ at 37 °C. The cell lines were authenticated by STR profiling, and mycoplasma testing was performed monthly with a MycoBlue Mycoplasma Detector (Vazyme, D101-01).

For transient protein expression, the cells were transfected with the indicated constructs via NEOFECT DNA transfection reagent (NeoFect, TF20121201) according to the manufacturer's instructions. After 48 h of transfection, the cells were harvested for the following assays. To generate stable overexpression or knockout cell lines, appropriate constructs and two packaging plasmids psPAX2 and pMD2.G were cotransfected into HEK293T cells. After 60 h, the lentivirus particles from the culture medium were collected and filtered with 0.45 μm filter, and then added to the host cells along with 8 μg/mL polybrene (Solarbio, H8761). Infected cells were then selected in puromycin (Solarbio, P8230)-containing medium. The protein levels of the target genes were determined via western blotting with the corresponding antibodies. To generate the double-knockdown cells, we selected a GFP-labeled KDM5C knockdown plasmid (pLKO.1-GFP) to infect cells depleted of YY1, and then screened the cells for puromycin resistance and green fluorescence. To restore KDM5C or H514A in knockdown cells, we inserted a hygromycin resistance gene into pHAGE-KDM5C/H514A-puro, which could be used to screen cells expressing KDM5C/H514A. The cells involved in the cell cycle and apoptosis assays did not contain fluorescence, which might have affected the results. We confirmed protein expression via western blotting.

## RNA extraction and qRT-PCR

Total RNA was extracted using the RNA isolater Total RNA Extraction Reagent (Vazyme, R401-01) according to the manufacturer's instructions. Reverse transcription reactions were performed by using ReverTra Ace qPCR RT Master Mix with gDNA Remover (TOYOBO, FSQ-301). RT-PCR was performed with ChamQ SYBR qPCR Master Mix (Vazyme, Q311-02) on a CFX Connect™ Real-Time System (Bio-Rad). The housekeeping gene expression levels of β-actin or GAPDH were used for normalization. Each reaction was performed in triplicates. For quantification of gene expression, the $2^{-\triangle\triangle Ct}$ method was used. The primer sequences of indicated genes were as follows:

h-JAM3-RT
Forward: 5'-GAGACTCAGCCCTTTATCGC-3';
Reverse: 5'-TGTTGCCATCTTGCCTACTG-3';
h-KNTC1-RT

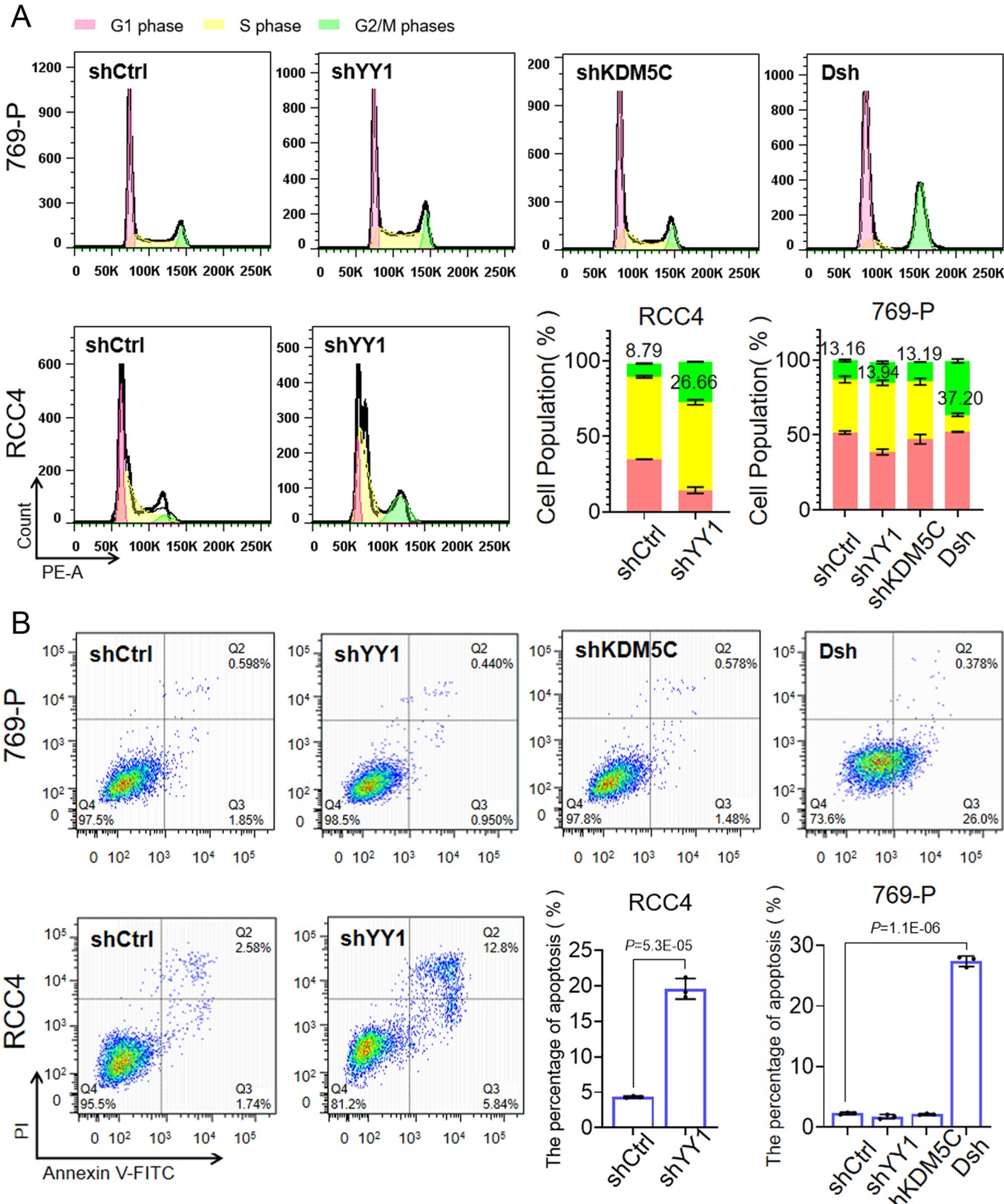

**Figure 6.  Simultaneous targeting of KDM5C and YY1 effectively induces cell cycle arrest in the G2/M phase and apoptosis.**

(**A**) Cell cycle analysis with the indicated cell lines via flow cytometry. The percentage of cells in each phase of the cell cycle is shown on the Y-axis. Each experiment was repeated three times with similar results. "Double sh" indicates cells with simultaneous knockdown of KDM5C and YY1. (**B**) Flow cytometry–based Annexin V/PI staining showing cells in the indicated lineages undergoing apoptosis. The quantitative results indicate the ratio of apoptotic cells. Each experiment was repeated three times with similar results. "Double sh" indicates cells with simultaneous knockdown of KDM5C and YY1. The data are shown as the mean ± SD of three independent experiments after analysis by two-tailed Student's $T$ test. $P = 5.3E\text{-}05$ (RCC4) and $P = 1.1E\text{-}06$ (769-P). Source data are available online for this figure.

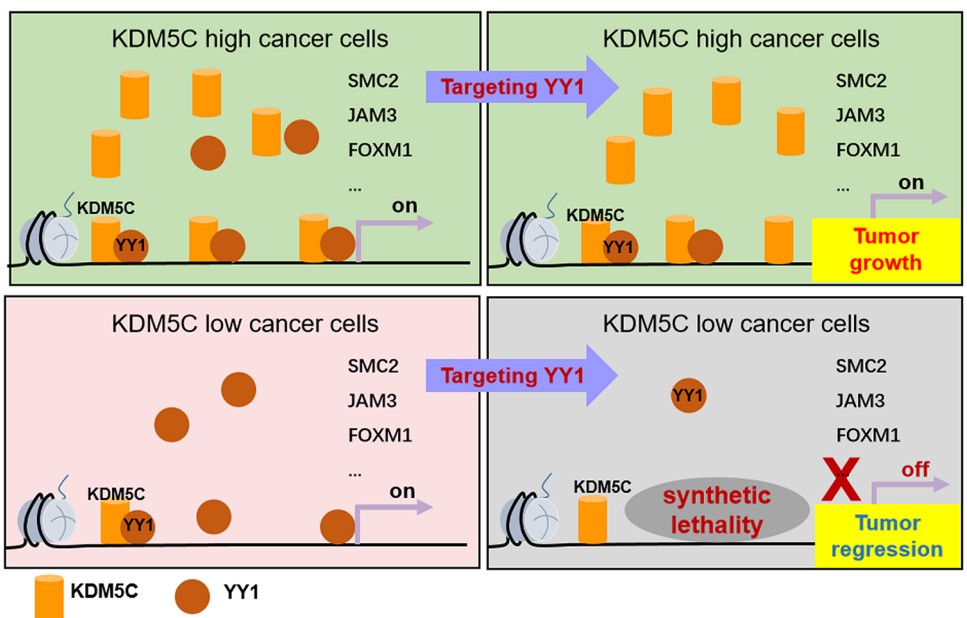

**Figure 7.  A mechanistic hypothesis model for KDM5C-mediated YY1 recruitment.**

YY1 and KDM5C constitute a pair of synthetic lethal genes. KDM5C recruits YY1 to the promoter regions of cell cycle- or apoptosis-related genes. Targeting YY1 increases the vulnerability of KDM5C-deficient cancer cells.

Forward: 5'-CTCACGAAGTTGCACAGG-3';
Reverse: 5'-GCCTCCAGCTCTTGTTTG-3';
h-SMC2-RT
Forward: 5'-GTTTTGGATGGTCTGGAGTT-3';
Reverse: 5'-ATTGGAGCAGGTTTGAAGAG-3';
h-SMC4-RT
Forward: 5'-TCGACCACCTAAGAAAAGTTG-3';
Reverse: 5'-CCATGAAGTAAAGGGGAGTG-3';
h-LMNB1-RT
Forward: 5'-CTGTCTCCAAGCCCTTCTT-3';
Reverse: 5'-ACTCGCCTCTGATTCTTCC-3';
h-FOXM1-RT
Forward: 5'-GCTGCCCCTTCCTGTTC-3';
Reverse: 5'-TGGACTCTGCCACTTCCTT-3';
h-GAPDH-RT
Forward: 5'-ACATCAAGAAGGTGGTGAAG-3';
Reverse: 5'-CTGTTGCTGTAGCCAAATTC-3'.

## Western blotting

The cell pellets were lysed with RIPA buffer (Beyotime, P0013B) containing a protease inhibitor mixture cocktail (MedChemExpress, HY-K0010). Then, the cells were centrifuged at 4 °C at 13,000 rpm for 10 min to remove insoluble cellular components. Total protein concentrations were measured with a BCA protein assay kit (Beyotime, P0012). Proteins (20–100 μg) were separated via 8–15% sodium dodecyl sulfate-polyacrylamide gel electrophoresis (SDS-PAGE) and transferred to nitrocellulose membranes (NC, PALL, 66485) in Tris-glycine buffer. The blots were blocked with 5% nonfat milk then incubated with diluted primary antibodies at 4 °C overnight. Then, appropriate horseradish peroxidase (HRP)-conjugated secondary antibodies were incubated with the blots for 1 h at room temperature. The signals were visualized with an enhanced chemiluminescence detection kit (ECL, Vazyme, E411-05) and exposed to X-ray film (Sigma, America). The primary antibodies used were purchased from commercial sources: anti-KDM5C (Cell Signaling Technology, D29B9; BETHYL, A301-034A; Proteintech, 14426-1-AP), anti-YY1 (Cell Signaling Technology, D5D9Z), anti-Flag (Proteintech, 20543-1-AP), anti-HA (Proteintech, 66006-2-Ig), anti-GAPDH (Proteintech, 60004-1-Ig), anti-β-actin (Proteintech, 66009-1-Ig), anti-H3 (Cell Signaling Technology, 2650) and anti-H3K4me3 (Cell Signaling Technology, 9751).

## Coimmunoprecipitation (Co-IP)

For the immunoprecipitation assay, appropriately constructed plasmids were transfected into the HEK293T and 769-P cell lines.

After 48 h of transfection, the cells were harvested and then lysed in IP lysis buffer (Beyotime, P0013) with EDTA-free protease inhibitor cocktail (MedChemExpress, HY-K0010) for 30 min on ice, sonicated, and then insoluble proteins were removed by centrifugation at 4 °C at 12,000 rpm for 10 min. An aliquot of the cleared cell lysate was collected as an input control. The remaining lysates were subsequently incubated with anti-HA or anti-Flag magnetic beads (MedChemExpress, HY-K0201, HY-K0207) overnight at 4 °C with gentle shaking.

For endogenous protein interactions, the cell lysates were processed as described above. One-tenth of each lysates was used as the input, half of the remainder was incubated with KDM5C antibody, and the other half was incubated with homologous IgG for 6 h at 4 °C. The lysates were then mixed with ProteinA/G agarose and incubated overnight at 4 °C.

Magnetic beads or agarose beads with a complex of antigen and antibody were washed four times in TBS-T buffer (100 nM Tris-HCl, 0.9% NaCl, 0.1% Tween-20). The proteins pulled down by the specified antibodies were eluted with 2×SDS loading buffer through boiling the beads, and then verified by western blotting.

## Tandem affinity purification and mass spectrometry (TAP-MS)

HEK293T cells overexpressing SBP-Flag-KDM5C were harvested for Tap-MS. Cell pellets were lysed with IP lysis buffer (Beyotime, P0013), and the supernatant was subsequently incubated with strep beads overnight at 4 °C. After three washes, SBP-linked protein were eluted with biotin solution, and then incubated with anti-Flag magnetic beads (MedChemExpress, HY-K0207). After enrichment of the target proteins, we washed the beads to remove nonspecific proteins and then performed SDS-PAGE on the products of the two groups. All of the lanes of the gel were collected for mass spectrometry analysis.

## Proliferation ability assay (CCK8)

The cells were seeded in a 96-well plate at 30–50% confluence according to the cell growth rate. When the cells just attached at the initial point, the OD450 nm was read every 24 h via a cell counting kit 8 (CCK8, MedChemExpress, HY-K0301) according to the manufacturer's instruction. The CCK8 solution was diluted with complete medium, incubated with the cells for 1–2 h, and then read with a microplate reader (BioTek Epoch, America) at 450 nm. The cell viability assay was performed in triplicate and repeated three times. Cell growth curves were drawn and analyzed via GraphPad Prism software V9.4.1.

## Colony formation assay

The cells were seeded into a six-well plate at the same cell density. Each well contained 1000 cells, which were cultivated in a cell incubator. Fresh cell culture medium was added every three days. After 4 weeks (ACHN and 769-P) or 8 weeks (RCC4), the cell culture medium was removed. The cells were gently washed with PBS, fixed with 4% paraformaldehyde for 20 min at 37 °C, and stained with 0.1% crystal violet in PBS. The assay was performed in triplicate and repeated two or three times.

## Cell cycle assay

The cells were gently collected at the exponential growth phase, washed with cold PBS, and then resuspended in 70% ethanol for more than 30 min at 4 °C. Then, the cells were washed with cold PBS, incubated with propidium iodide (PI) solution (Byotime, C1052) and protected from light at 37 °C for 30 min. Red fluorescence was detected via flow cytometry (LSRFortessaX20, BD) at Ex=488 nm, and then analyzed with Flowjo. The assay was repeated two or three times.

## Cell apoptosis assay

In accordance with the instructions of the Annexin V-FITC/PI apoptosis kit (Multi Sciences, AP101), we collected $1–10 × 10^5$ cells and resuspended them in 500 μl of 1× Binding buffer, and add 5 μl of Annexin V-FITC and 10 μl of PI to each sample. The samples were mixed gently and protected from light for 5 min, then Annexin V-FITC (Ex = 488 nm; Em = 530 nm) and PI (Ex = 535 nm; Em = 615 nm) were detected via flow cytometry (LSRFortessaX20, BD) with. The results were analyzed via Flowjo.

## Isolation of chromatin-bound proteins from subcellular fractions

After the transfection of HEK293T cells with empty vector, Flag-KDM5C or Flag-H514A, we collected the cells and lysed them with E1 buffer (50 nM HEPES-KOH, 140 nM NaCl, 1 mM EDTA, 10% glycerol, 0.5% NP-40, 0.25% TritonX-100, 1 mM DTT, Protease inhibitor cocktail). The cells were subsequently centrifuged $1100×g$ at 4 °C for 2 min after which the supernatant was obtained as the cytoplasmic fraction. Next, we resuspended the pellet in E2 buffer (10 nM Tris-HCl, 200 mM NaCl, 1 mM EDTA, 0.5 mM EGTA, Protease inhibitor cocktail). After centrifugation, the supernatant was obtained as the nuclear fraction. We resuspended the pellet in E3 buffer (500 mM Tris-HCl, 500 mM NaCl, protease inhibitor cocktail) and sonicated it on ice for 5 min, 2 s ON/45 s OFF. The supernatant was centrifuged at 4 °C for 10 min at $16,000×g$, and transferred to fresh microcentrifuge tubes. The protein concentrations were analyzed via western blotting.

## Chromatin immunoprecipitation followed by next-generation sequencing (ChIP-seq)

The cells were plated in 10 cm dishes until they reached confluence and were then fixed with 37% fresh formaldehyde to a final concentration of 1% for 13 min at room temperature with rotation, after which the reaction was quenched with 0.125 mM glycine for 10 min. The final number of Chip-seq cells in each group was $4 × 10^7$.

The cells were washed three times with ice-cold PBS and then harvested with a cell scraper and lysed in lysis buffer 1 (50 mM HEPES-KOH, 140 mM NaCl, 1 mM EDTA, 10% glycerin, 0.5% NP-40, 0.25% TritonX-100), lysis buffer 2 (pH 8.0 10 mM Tris-HCl, 200 mM NaCl 1 mM EDTA, 0.5 mM EGTA), and lysis buffer 3 (pH 8.0 10 mM Tris-HCl, 100 mM NaCl, 1 mM EDTA, 0.5 mM EGTA, 0.2% N-lauroyl sarcosine sodium salt). The chromatin was sonicated in lysis buffer 3 to ensure that the size of the chromatin fraction was between 200 and 500 bp. In total, 80 μL of each sample was transferred to a clean tube as the input DNA. The chromatin sample was diluted with ChIP dilution buffer (20 mM Tris-HCl, 150 mM NaCl, 1 mM EDTA, 1% TritonX-100).

Overall, 6 μL of antibody was added to each tube and the mixture was incubated at 4 °C overnight. To each tube, 80 μL of protein G Sepharose 4 Fast Flow (GE Healthcare, 17061805) was added, and the immune complexes were collected by incubation at 4 °C for at least 6 h with gentle rotation. The bound complexes were washed once with each of the following buffers: low-salt wash buffer (0.1% SDS, 1% TritonX-100, 2 mM EDTA, 20 mM Tris-HCl pH 8.0, 150 mM NaCl), high-salt wash buffer (0.1% SDS, 1% TritonX-100, 2 mM EDTA, 20 mM Tris-HCl pH 8.0, 500 mM NaCl), LiCl wash buffer (0.25 M LiCl, 1% NP-40, 1 mM EDTA, 10 mM Tris-HCl pH 8.0) and TE buffer (10 mM Tris-HCl pH 8.0, 1 mM EDTA). The beads were incubated at 65 °C with TE buffer containing proteinase K, RNase and SDS overnight, and the DNA was extracted using phenol/chloroform. ChIP-seq was performed by Beijing Novogene Biotech Co., Ltd.

## ChIP-qPCR

The qPCR templates were obtained from ChIP. For method details, please refer to "qRT-PCR". We calculated the fold enrichment of those genes via YY1-ChIP, and IgG was used as a negative control. The calculation formula is as follows:

$$\Delta Ct[\text{normalized ChIP}] = (Ct[\text{ChIP}] - (Ct[\text{Input}] - \text{Log2}(\text{Input Dilution Factor})))$$

$$\Delta\Delta Ct[\text{ChIP/NIS}] = \Delta Ct[\text{normalized ChIP}] - \Delta Ct[\text{IgG}]$$

$$\text{Fold Enrichment} = 2^{(-\Delta\Delta Ct[\text{ChIP/NIS}])}$$

Primer sequences used in ChIP-qPCR are as follows:
JAM3
Forward: 5'-TGGTAACTGGGGCGGGTCGCA-3';
Reverse: 5'-AGCTTCGGCCCGCTCTGCTTC-3';
KNTC1
Forward: 5'-CAACACCGCCTTACTTCC-3';
Reverse: 5'-CCGGGACTCTGAACTATGGC-3';
SMC2
Forward: 5'-GGAGCTACCATTATCGAGAC-3';
Reverse: 5'-CTTGACGCACCCCAAAAG-3';
SMC4
Forward: 5'-AAAAGAATCCCTCGCTCTTC-3';
Reverse: 5'-TGACAGCAAGAGAAATCTCC-3';
LMNB1
Forward: 5'-TTGCCCAAGGGCCAGATTTTA-3';
Reverse: 5'-AATGCAAGACGAGGGTGACT-3';
FOXM1
Forward: 5'-ATTTTCCCACAGTGAACGAT;
Reverse: 5'-AACCTTGTCTGCCATTGTATC.

## Cleavage under targets and tagmentation assay

The Cleavage Under Targets and Tagmentation (CUT&Tag) assay was conducted according to the manual instructions of Hyperactive Universal CUT&Tag Assay Kit for Illumina Pro (Vazyme, TD904). A ChIP-grade anti-KDM5C antibody (Bethyl, A301-034A) and anti-YY1 (CST, D5D9Z) were used for immunoprecipitation. Rabbit IgG (ABclonal, AC042) was used as a negative control.

The library was constructed with the TruePrep Index Kit V2 for Illumina (Vazyme, TD202).

## ChIP-seq and CUT&Tag data analysis

The raw data (raw reads) in fastq format were first processed via fastp software to obtain clean data, from which reads containing adapters were removed. The reference genome hg38 and gene model annotation files were downloaded directly from the genome website. The index of the reference genome was built using Bowtie2, and the clean reads were aligned to the reference genome with Bowtie2. After the reads were mapped to the reference genome, MACS2 peak-calling software was used to identify regions enriched with IP over the background. A q-value threshold of 0.05 was used for all of the datasets. Only peaks with counts greater than 200 were retained in the CUT&Tag data. Bedtools reldist and bedtools intersect were used to compare two Chip-seq or CUT&Tag datasets in the global or promoter region. Motif analysis was performed via Homer. Chipseeker was used to analyze genomic distribution.

## RNA-seq and data analysis

Total RNA was extracted from the corresponding cells. The concentration was measured by Qubit 3.0 fluorometer (Life Technologies) and RNA purity was checked via the kaiaoK5500®-Spectrophotometer (Kaiao, Beijing, China). Sample integrity was assessed using an Agilent 2100 RNA Nano 6000 assay kit (Agilent Technologies). RNA-seq was performed by Annoroad Gene Technology Corporation (Beijing).

Briefly, sequencing libraries were constructed using the NEBNext Ultra RNA Library Prep Kit for Illumina (NEB, USA). Clustering of the index-coded samples was performed on a cBot cluster generation system using the HiSeq PE Cluster Kit v4-cBot-HS (Illumina) according to the manufacturer's instructions. After cluster generation, the libraries were sequenced on an Illumina platform and 150 bp paired-end reads were generated.

For the data analysis, an index of the reference genome was built using HISAT2 v2.0.5 and paired-end clean reads were aligned to the reference genome via HISAT2 v2.0.5. FeatureCounts v1.5.0-p3 was used to count the numbers of reads mapped to each gene. Differential expression analysis of the two groups was performed using the DESeq2 R package. Heatmaps were generated via pheatmap (v1.0.12) package in R.

## Xenograft assay

All animal experiments were approved by the Animal Research Ethics Board of Wuhan University (WAEF-2022-0113) and confirmed by the animal experimental guidelines. Four-week-old female BALB/c nude mice were purchased from Beijing Vital River Laboratory Animal Technology (China) and were housed under specific pathogen-free conditions in Animal Experiment Center-Animal Biosafety Level-III Laboratory of Wuhan University. The animals were randomly divided into four groups of six mice each. The mice were injected with 100 μL of cells suspended in Matrigel Basement Membrane Matrix (Corning, 356234) at a population of $3 \times 10^6$ ACHN indicated cells into the left or right dorsal flank subcutaneously.

The tumor size was estimated once a week via a digital Vernier caliper. The tumor volume was calculated with the following formula: tumor volume $(mm^3)$ = length (mm) × width$^2$ (mm) × 0.5. Tumors were harvested 8 weeks after injection.

## Statistical analysis

The quantitative experiment was repeated three times or more independently. The graphs were generated with GraphPad Prism software. Two-tailed Student's $t$ test, one-way ANOVA and the Wilcoxon rank-sum test were used to assess the significance of the experiments. $P$ values < 0.05 were considered to indicate the significant differences.

## Data availability

The sequencing data are available in GEO Data Sets with the accession number GSE270718, GSE270719 and GSE275838. All other raw data support the findings of this study are available from the corresponding author upon reasonable request.

The source data of this paper are collected in the following database record: biostudies:S-SCDT-10_1038-S44319-024-00290-8.

## Peer review information

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

## Acknowledgements

This work was supported by Natural Science Foundation of China (grant numbers 82203491 and 82173137), Natural Science Foundation of Hubei Province of China (grant number 2022CFA008), China Postdoctoral Science Foundation (grant number 2022TQ0237), Natural Science Foundation of Inner Mongolia (grant number 2024QN08005) and Translational Medicine and Interdisciplinary Research Joint Fund of Zhongnan Hospital of Wuhan University (grant number ZNJC202212).

## Author contributions

**Qian Zheng**: Funding acquisition; Validation; Methodology; Writing—original draft. **Pengfei Li**: Conceptualization; Funding acquisition; Methodology. **Yulong Qiang**: Software; Formal analysis; Validation; Methodology; Project administration. **Jiachen Fan**: Visualization. **Yuzhu Xing**: Formal analysis. **Ying Zhang**: Visualization. **Fan Yang**: Formal analysis; Validation; Methodology; Writing—review and editing. **Feng Li**: Conceptualization; Data curation; Supervision; Funding acquisition; Investigation; Writing—original draft; Project administration; Writing—review and editing. **Jie Xiong**: Conceptualization; Supervision; Visualization; Writing—original draft; Project administration; Writing—review and editing.

Source data underlying figure panels in this paper may have individual authorship assigned. Where available, figure panel/source data authorship is listed in the following database record: biostudies:S-SCDT-10_1038-S44319-024-00290-8.

## Disclosure and competing interests statement

The authors declare no competing interests.

# Expanded View Figures

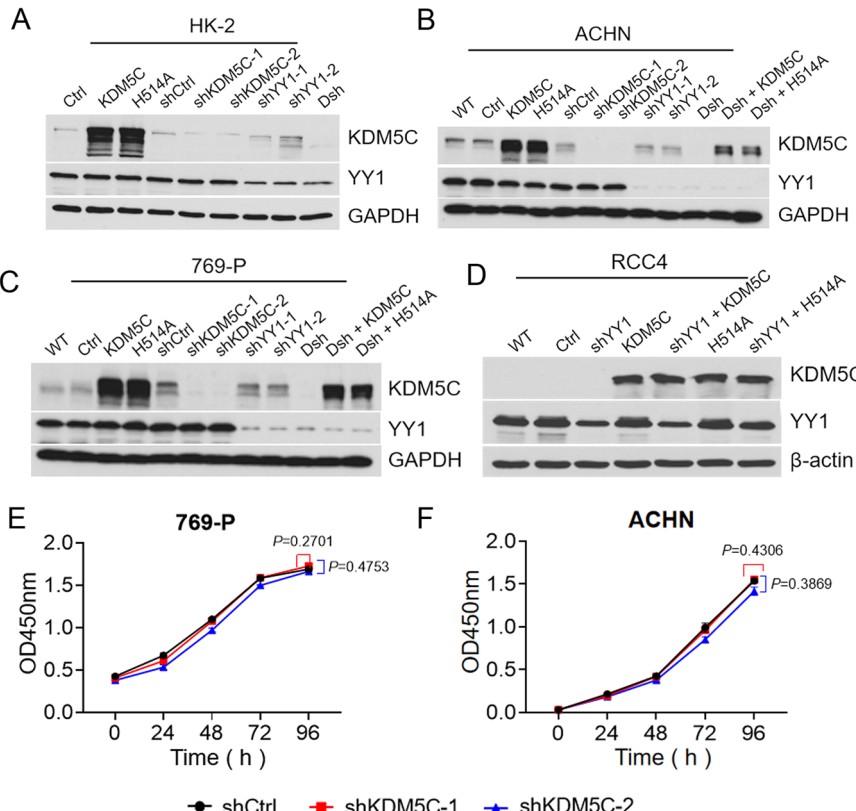

**Figure EV1. Validation of the constructed cell lines used in this study.**

(A–D) KDM5C and YY1 protein levels in the indicated cells were measured via western blotting. "Dsh" indicates cells in which KDM5C and YY1 were both knocked down. (E, F) Proliferation rate of the indicated cell lines upon KDM5C depletion, as determined via a CCK8 assay. The data are shown as the mean ± SD of three independent experiments after analysis via one-way ANOVA. In 769-P cells, $P = 0.2701$ (shKDM5C -1) and $P = 0.4753$ (shKDM5C-2); in ACHN cells, $P = 0.4306$ (shKDM5C-1) and $P = 0.3869$ (shYY1-2). Source data are available online for this figure.

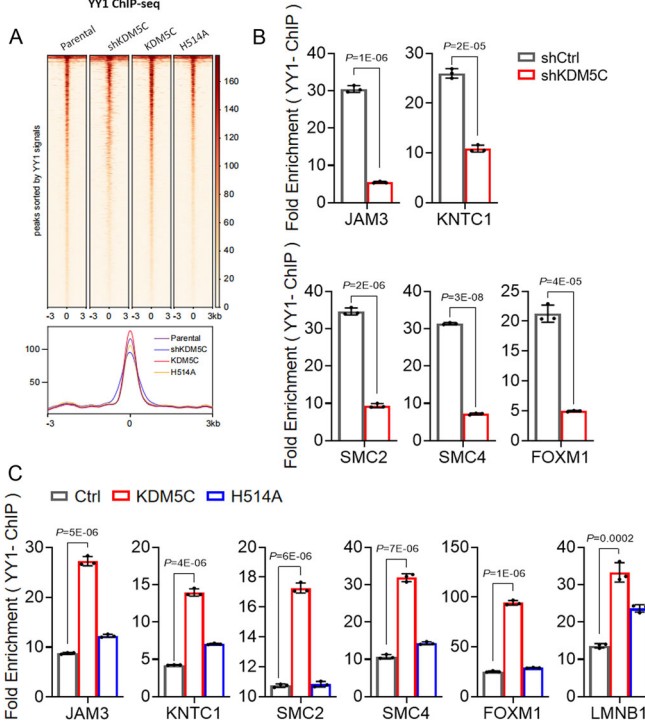

**Figure EV2. ChIP-seq experiments targeting YY1 in HK-2 cells.**

(A) Heatmap and metaplot showing the chromatin enrichment of YY1 in HK-2 cells in the presence or absence of KDM5C. (B, C) ChIP-PCR assay showing the enrichment of YY1 at the promoters of the indicated genes in HK-2 cells after the depletion of YY1, KDM5C or both (B), or in cells expressing KDM5C-WT or KDM5C-H514A mutant (C). Each experiment was repeated three times with similar results. Statistical significance was determined by one-way ANOVA. (B) $P = 1.24E\text{-}06$ (JAM3), $P = 2.27E\text{-}06$ (KNTC1), $P = 2.82E\text{-}08$ (SMC2), $P = 2.41E\text{-}05$ (SMC4) and $P = 3.96E\text{-}05$ (FOXM1), respectively. (C) $P = 4.53E\text{-}06$ (JAM3), $P = 4.46E\text{-}06$ (KNTC1), $P = 5.53E\text{-}06$(SMC2), $P = 6.82E\text{-}06$ (SMC4), $P = 1.05E\text{-}06$ (FOXM1) and $P = 0.0002$ (LMNB1), respectively. Source data are available online for this figure.

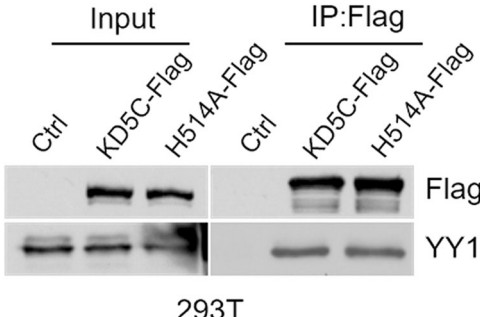

**Figure EV3. Evaluation of the interaction between YY1 and KDM5C-WT or KDM5C-H514A mutant.**

Co-IP followed by western blotting showing the interaction between YY1 and KDM5C-WT or KDM5C-H514A mutant in HEK293T cells. Source data are available online for this figure.

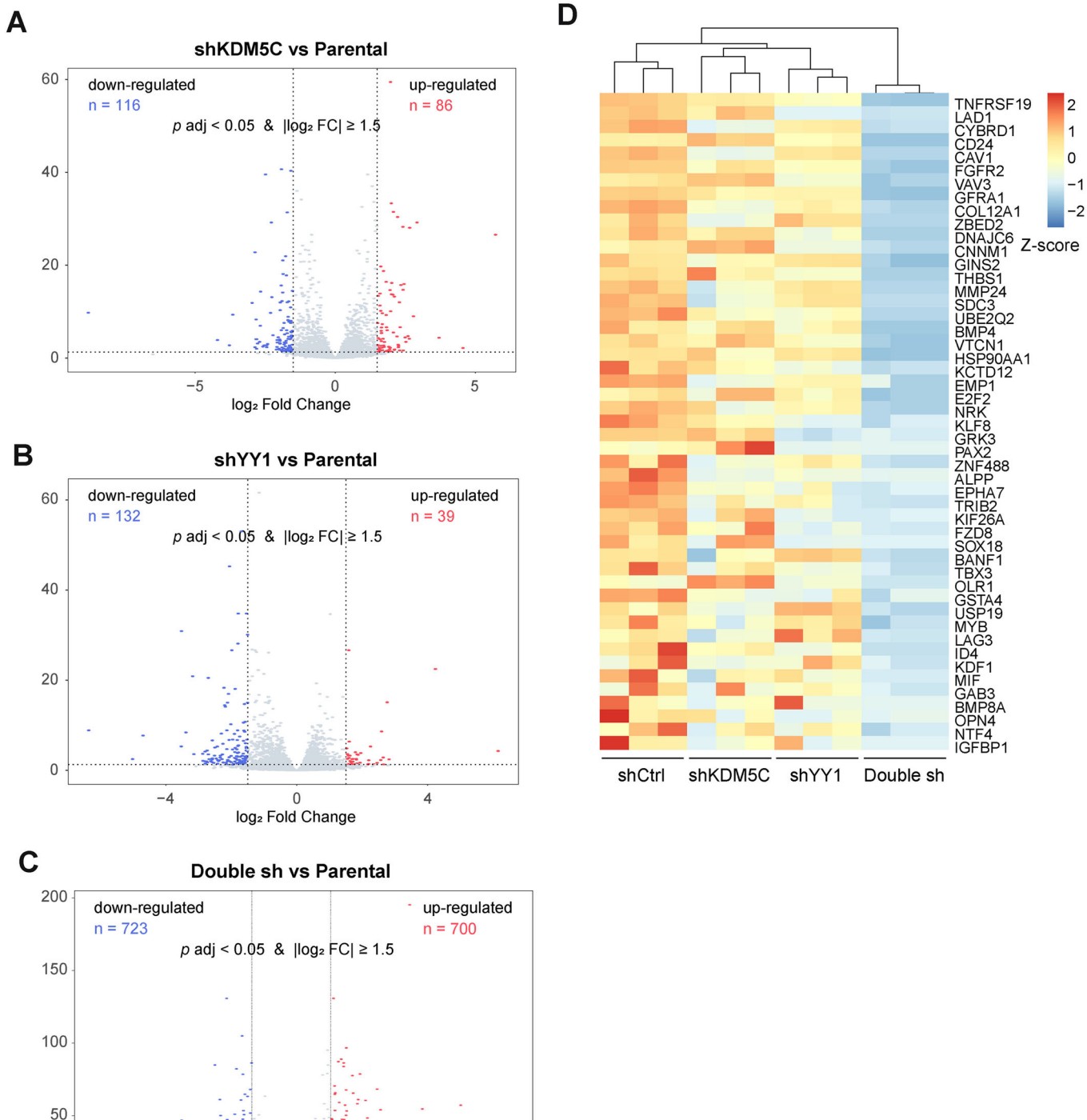

**Figure EV4. RNA-seq in HK-2 derived cells.**

(A–C) Volcano plots showing the change in the number of transcripts in shKDM5C, shYY1 or double-knockdown HK-2 cells, as determined by RNA-seq analysis ( | log$_2$ FC | > 1.5, P value < 0.05). (D) Heatmap showing the normalized gene expression profile in shKDM5C, shYY1 or double sh ACHN cells. "Double sh" indicates cells in which KDM5C and YY1 were both knocked down. Source data are available online for this figure.

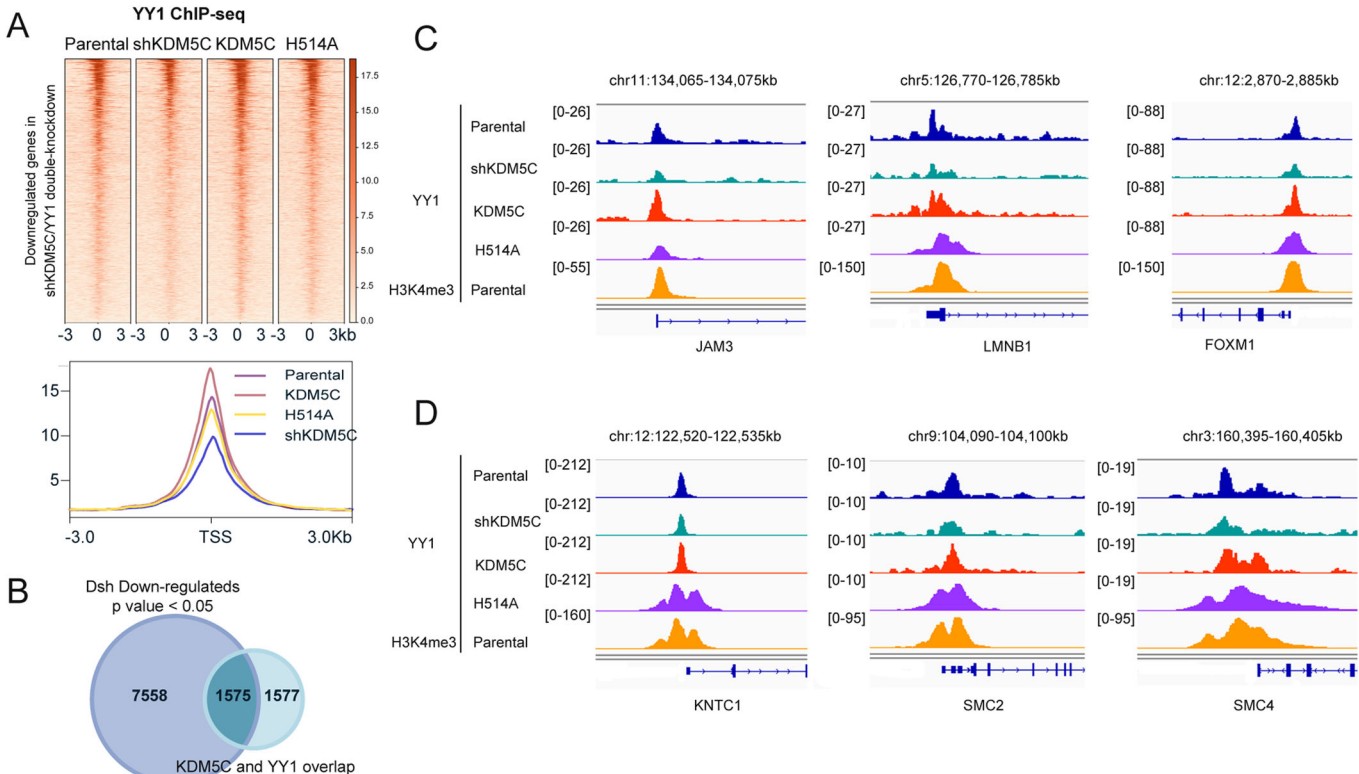

**Figure EV5. Effects of KDM5C on YY1 enrichment at promoter regions of specific genes.**

(A, B) The connections between the ChIP-seq and RNA-seq data were analyzed, and the genomic regions of the downregulated genes in the KDM5C/YY1 double-knockdown group were annotated. (C, D) ChIP-seq snapshots reflecting changes in YY1 enrichment at the promoter regions of the JAM3, LNMB1, FOXM1, KNTC1, SMC2 and SMC4 genes in the indicated cell lines. The promoters are indicated by H3K4me3 peaks. Source data are available online for this figure.

