## [Peer Review File · EMBO Reports]

Targeting the transcription factor YY1 is synthetic lethal with loss of the histone demethylase KDM5C

Qian Zheng, Pengfei Li, Yulong Qiang, Jiachen Fan, Yuzhu Xing, Ying Zhang, Fan Yang, Feng Li, and Jie Xiong

Corresponding author(s): Jie Xiong (jjexiong@whu.edu.cn) , Feng Li (fli222@whu.edu.cn), Fan Yang (yangfan2022@whu.edu.cn)

Review Timeline:

Submission Date:	2nd Jan 24
Editorial Decision:	25th Jan 24
Revision Received:	14th Jun 24
Editorial Decision:	18th Jul 24
Revision Received:	4th Sep 24
Editorial Decision:	12th Sep 24
Revision Received:	25th Sep 24
Editorial Decision:	4th Oct 24
Revision Received:	4th Oct 24
Accepted:	8th Oct 24

Editor: Esther Schnapp

Transaction Report:

Dear Mrs. Xiong,

Thank you for the submission of your manuscript to EMBO reports. We have now received the full set of referee reports that is pasted below.

As you will see, the referees acknowledge that the findings are potentially interesting. However, they also raise several concerns and point out that significant revisions will be required before the work can be considered for publication here. I think all points raised are reasonable and should be addressed, except for point 3 by referee 1, which does not need to be addressed experimentally for the publication of your work here. One of the main concerns seems to be the inconsistent use of cell lines, but only referee 2 recommends to focus the study on one type of cancer. I would like to suggest that you send us a proposed revision plan that I can share with the referees before you embark on these extensive revisions. This should help to clarify the exact revision requirements. Please let me know if you have any questions or comments.

I would thus like to invite you to revise your manuscript with the understanding that the referee concerns must be fully addressed and their suggestions taken on board. Please address all referee concerns in a complete point-by-point response. Acceptance of the manuscript will depend on a positive outcome of a second round of review. It is EMBO reports policy to allow a single round of major revision only and acceptance or rejection of the manuscript will therefore depend on the completeness of your responses included in the next, final version of the manuscript.

We realize that it is difficult to revise to a specific deadline. In the interest of protecting the conceptual advance provided by the work, we recommend a revision within 3 months (26th Apr 2024). Please discuss the revision progress ahead of this time with the editor if you require more time to complete the revisions.

- 1) A data availability section providing access to data deposited in public databases is missing. If you have not deposited any data, please add a sentence to the data availability section that explains that.
- 2) Your manuscript contains statistics and error bars based on $n=2$. Please use scatter blots in these cases. No statistics should be calculated if $n=2$.

5) a complete author checklist, which you can download from our author guidelines <https://www.embopress.org/page/journal/14693178/authorguide>. Please insert information in the checklist that is also reflected in the manuscript. The completed author checklist will also be part of the RPF.

6) Please note that all corresponding authors are required to supply an ORCID ID for their name upon submission of a revised manuscript (<https://orcid.org/>). Please find instructions on how to link your ORCID ID to your account in our manuscript

tracking system in our Author guidelines

<<https://www.embopress.org/page/journal/14693178/authorguide#authorshippingguidelines>>

10) Regarding data quantification (see Figure Legends:

<https://www.embopress.org/page/journal/14693178/authorguide#figureformat>)

I look forward to seeing a revised form of your manuscript when it is ready.

Yours sincerely,

Referee #1:

Zheng et al., presented a study on identifying YY1 as a potential synthetic lethality target for KDM5C in cancer cells. Mechanistically, KDM5C is important for global YY1 chromatin recruitment. Functionally, KDM5C and YY1 combined KO caused better anti-tumor efficacy. Overall, this manuscript provides some novel insight on the synthetic lethality interaction between YY1 and KDM5C. On the other hand, a few key weaknesses outlined below should be addressed.

1. The data quality in Fig.1f needs to be improved. In addition, it is best to perform this experiment with semi-endogenous IP.
2. Fig.2, these functional experiments need to be confirmed with KDM5C isogenic cell lines, which is critical. This also applies to the following figures in the manuscript, which need to have isogenic cell lines in place to validate the claim.
3. It is unclear what downstream targets that may contribute to the synthetic lethality, which needs to be explored further.

Referee #2:

In the paper by Zheng et al., the authors propose a mechanistic crosstalk between the demethylase KDM5C and the transcription factor YY1 that exhibits a synergistic antitumor effect when both are depleted. The newly identified role of the scaffolding function of KDM5C is intriguing, and the work is well conducted, with some of the proposed hypotheses being adequately addressed. However, several important points need clarification and addressing:

- 1- A crucial question for this reviewer concerns the extent to which the roles of KDM5C and YY1 proposed in this manuscript constitute unique characteristics of tumor cells. The ChIP-seq of YY1 in wild-type and KDM5C-depleted non-tumoral cells should be performed and compared with data presented in Figure 3C (see also point 2). Additionally, the analysis of double knockdown of YY1 and KDM5C by RNA-seq in a non-tumoral cell context and compared with data presented in Figure 5C, will also help to address this point (see point 2), especially considering the diverse roles that KDM5C may have depending on the type of cancer (pages 3-4).
- 2- Throughout the paper, the cell model is changed (293T, ACHN, RCC4 cells). If feasible, it is advisable to concentrate the manuscript on a specific type of cancer rather than too many cell types, where the proposed mechanism may not be conserved. This focused approach will enhance the ability to unravel the function of the YY1-KDM5C complex within a more defined context and will contribute to drawing more solid conclusions. Thus, I recommend that the authors narrow their study to one type of cancer (renal? cervical?) and use the normal counterpart as previously suggested (point 1)
- 3- In Figure 3C, among the 4928 regions where YY1 binding decreases in the absence of KDM5C, i) what about YY1 levels in regions that do not overlap with KDM5C (9473 regions)? Are these regions not affected by the knockdown?
ii) Considering the authors' proposition that YY1 binding depends on KDM5C and given YY1's ability to bind DNA to a specific DNA sequence, are the regions where both KDM5C and YY1 overlap enriched in YY1 DNA binding motifs? Alternatively, could they be enriched in another DNA binding motif or transcription factor (TF) that could serve as a connecting element between KDM5C and YY1? How would this crosstalk occur mechanistically?
iii) The authors should link ChIP-seq (Figure 4) with RNA-seq results (Figure 5). Are genes affected by knockdowns significantly enriched in YY1 and KDM5C at the promoters?
- 4- The KDM5C ChIP-seq data are obtained from GSE71327, primarily performed on ZR-75-30, an epithelial cell line isolated from the mammary gland of a ductal carcinoma patient. On which cells have the experiments included in the manuscript been performed? Is there any reason not to utilize data from a system that is more compatible with the study in question?
- 5- In Figure 4A, the authors conclude that "ectopic KDM5C enhances YY1 chromatin binding," but the data presented show a very weak effect (despite the strong overexpression of KDM5C, Fig S2A). Could the authors clarify this point?
- 6- In Figure 5, the cell model used for RNA-seq analysis is not mentioned. This information is vital as the authors aim to connect the expression results with the functional data presented in Figure 2.

Minor points:

- 1- In Figure 2A, the levels of YY1 should be displayed.
- 2- Figure 6 should be moved to Supplementary.
- 3- In Figure 7C, the model is unclear, and the authors should revise it to provide a more straightforward and comprehensible representation that succinctly summarizes the conclusions of their work.

Referee #3:

In this work, the authors presented data showing a synthetic lethal interaction between YY1 and KDM5C and the inhibition of both could be a potential combination therapy for cancer treatment. They showed that YY1 knockdown only affected tumor cell

proliferation when KDM5C expression was low or knocked down. Mechanistically, KDM5C can recruit YY1 to the promoters of several cell-fate essential genes and regulate their expression. The findings are interesting and novel. However, the overall writing needs to be improved and some points are not adequately supported. Moreover, the major concern I have is that the cell line used for mechanistic study is not one of the cell lines used for phenotypic characterization.

Major points:

1. In Fig1, the authors showed the interaction between KDM5C and YY1 in 293T cells exogenously and endogenously. Is this interaction also present in CAA3 and 769P cells? In Fig1F in the input blot, why is the background in each lane different? Some of the lanes even share some common bands. To address this, a control with an empty vector needs to be included.
2. In Fig3, the authors used ChIP-seq to explore the underlying mechanism. Why did the authors choose to use ACHN cell line? The authors need to show the phenotype in ACHN caused by KDM5C and YY1 depletion or conduct ChIP-seq in CAA3 and 769P cells. Similarly in Fig3B, what cell line was used in KDM5C ChIP-seq? The cell lines used need to be consistent throughout the study.
3. In Fig4, the authors focused a lot on the H514A mutant of KDM5C. The authors showed that H514A failed to increase YY1 recruitment like WT KDM5C and H3K4me3 does not affect YY1 recruitment. However, H514A can still interact with YY1 and it can partially rescue the phenotype in RRC4 but not in CaSki, which is very confusing. Could the authors put KDM5C-WT or KDM5C-H514A back to KDM5C and YY1 double knockdown CAA3 and 769P cells to clarify this? To explain this, the authors claimed that H514A "influenced the chromatin binding activity of KDM5C". This claim was only backed up by Fig4F, which is not sufficient, and it goes against a published study(PMID: 33977073). Much work needs to be done to support this claim.
4. Again, the cell lines used for RNA-seq analysis is not specified. What pathways are enriched upon KDM5C and YY1 double knockdown? Also, there is an inconsistency between the qPCR and RNA-seq data. In the RNA-seq result, SMC2 and SMC4 is the highest in shYY1 group and LMNB is the lowest in shKDM5C group. However, the qPCR results showed a different trend. Are these genes also downregulated in double knockout CAA3 and ACHN cells?
5. In Fig6A-B and FigS3, the KDM5C-H514A group seems to be left out. Did KDM5C H514A change the size of YY1 peaks? Also the authors need to verify these results using ChIP-qPCR. To add to the robustness of the data, the authors could also perform ChIP-qPCR in different cell lines.

Minor points:

1. The figure legend of 2D and 2E was switched.
2. The western blot images could be cropped taller generally. Some bands are cropped out.
3. In page 7, it says "the intensity of YY1 binding peaks in both KDM5C-depleted and parental ACHN was reduced upon loss of KDM5C". Do the authors mean "the intensity of YY1 binding peaks was reduced upon loss of KDM5C"?
4. 769-P cell line is sometimes labeled 769P.
5. More details are generally needed in the method section. Some coIP experiments and ChIP experiments weren't included. ACHN KDM5C-H514A was left out in the ChIP method.

Dear Dr. Schnapp,

On behalf of my co-authors and myself, I would like to thank you and other editors of *EMBO Reports* for providing us the opportunity to revise the manuscript. We greatly appreciate all the constructive suggestion and comments on our manuscript (EMBOR-2024-58727V1). **We have prepared the revised edition according to the reviewers' suggestion.** Attached please find our revised manuscript. **Below is our response (in blue) to the reviewers' comments (in black).** Please feel free to contact me if you have further questions or need additional information.

Referee #1,

Zheng et al., presented a study on identifying YY1 as a potential synthetic lethality target for KDM5C in cancer cells. Mechanistically, KDM5C is important for global YY1 chromatin recruitment. Functionally, KDM5C and YY1 combined KO caused better anti-tumor efficacy. Overall, this manuscript provides some novel insight on the synthetic lethality interaction between YY1 and KDM5C. On the other hand, a few key weaknesses outlined below should be addressed.

1. The data quality in Fig.1f needs to improved. In addition, it is best to perform this experiment with semi-endogenous IP.

Response: We appreciate your comments. To improve the quality of Fig.1f, we performed semi-endogenous co-IP between KDM5C truncation mutants and full-length YY1, and adding an empty vector control, which was shown as Fig 1G in the revised manuscript. In Fig 1C and 1D, we have shown the interaction between KDM5C and YY1 in 293T cells exogenously and endogenously. According to your suggestion, we then performed the semi-endogenous co-IP with YY1 and KDM5C-Flag in 769-P renal cancer cell lines, as shown in Fig 1E, KDM5C clearly bound to YY1.

2. Fig.2, these functional experiments need to be confirmed with KDM5C isogenic cell lines, which is critical. This also applies to the following figures in the manuscript, which needs to have isogenic cell lines in place to validate the claim.

Response: According to the reviewers' suggestion, we mainly focused on the renal cancer cell lines in the revised manuscript. Additional experiments with isogenic cell lines were performed, including clonal formation (Fig 2B), CCK-8 (Fig 2C), and xenograft assay (Fig 2D). Besides, the effect of KDM5C or H514A mutant re-expression in YY1 depleted or double knockdown isogenic cell lines were also examined (4G-J). Our results collectively confirmed that YY1 is a vulnerable target in KDM5C-deficient tumor cells.

3. It is unclear what downstream targets that may contribute to the synthetic lethality, which needs to be explored further.

Response: We appreciate your comments. Double knockdown of KDM5C and YY1 but not loss of either of them significantly suppressed a number of essential cell fate genes, including JAM3, KNTC1, SMC2, SMC4, LMNB1 and FOXM1. The related results were presented in Fig 5-6.

Those genes play important roles in modulating cell proliferation and cell cycle. High JAM3 expression is correlated with downregulation of GSK3 β and activation of β -catenin/CCND1 signaling, which is sufficient to maintain the self-renewal ability and cell cycle^[1]. Kinetochores associated 1 (KNTC1) was highly expressed in several carcinoma tissues and correlated with poor prognosis. KNTC1 knockdown leads to inhibited proliferation and migration, as well as G2 phase arrest and enhanced apoptosis^[2]. The SMC family includes condensin and cohesin, which structure chromosomes to enable mitosis and long-range gene regulation. Human SMC2 is responsible for tightly packaging replicated genomic DNA prior to segregation into daughter cells. WNT signaling can directly activate SMC2 transcription as a key player in the mitotic cell division machinery^[3]. SMC4 has been reported to be involved in tumor cell growth, migration and invasion, and to be correlated with poor prognosis of cancer patients^[4]. LMNB1 promotes the progression of hepatocellular carcinoma progression by regulating the PI3K and MAPK pathways, serving as an effective therapeutic target as well as a reliable prognostic biomarker for HCC^[5]. FOXM1 is frequently overexpressed in a variety of human cancers. It plays an oncogenic role by interacting with other proteins, such as β -catenin or SMAD3 to induce oncogenic WNT and TGF β signaling pathways, respectively^[6].

Referee #2:

In the paper by Zheng et al., the authors propose a mechanistic crosstalk between the demethylase KDM5C and the transcription factor YY1 that exhibits a synergistic antitumor effect when both are depleted. The newly identified role of the scaffolding function of KDM5C is intriguing, and the work is well conducted, with some of the proposed hypotheses being adequately addressed. However, several important points need clarification and addressing:

1. A crucial question for this reviewer concerns the extent to which the roles of KDM5C and YY1 proposed in this manuscript constitute unique characteristics of tumor cells. The ChIP-seq of YY1 in wild-type and KDM5C-depleted non-tumoral cells should be performed and compared with data presented in Figure 3C (see also point 2). Additionally, the analysis of double knockdown of YY1 and KDM5C by RNA-seq in a non-tumoral cell context and compared with data presented in Figure 5C, will also help to address this point (see point 2), especially considering the diverse roles that KDM5C may have depending on the type of cancer (pages 3-4).

Response: We sincerely thank the reviewer for the constructive suggestion. We additionally performed the ChIP-seq experiments targeting YY1 in HK-2 cell line

(human kidney proximal tubular epithelial cell line) in the presence or absence of KDM5C (Fig EV2A). The results revealed that the non-tumoral cell HK-2 and renal cancer cell ACHN showed similar pattern, namely, the intensity of YY1 binding peaks was partially relying on the KDM5C protein level, suggesting that KDM5C mediated YY1 chromatin recruitment may represent a universal mechanism of action

We also performed RNA-seq in HK-2 derived cells. As shown in Fig EV4A-B, loss of KDM5C or YY1 led to the upregulated or downregulated expression of a number of genes. Consistently, double knockdown results in more downregulated genes in HK-2 cells, although to a lesser extent compared to that in ACHN cells (Fig EV4C). Importantly, simultaneous loss of KDM5C and YY1 but not one of them also significantly suppresses HK-2 cell proliferation (Fig R1).

Taken together, our work demonstrates a synthetic lethal interaction between YY1 and KDM5C both in tumor and non-tumoral cells.

Fig R1. HK-2 cells proliferation after shKDM5C, shYY1 and double knockdown, as measured by CCK8 assay.

2. Throughout the paper, the cell model is changed (293T, ACHN, RCC4 cells). If feasible, it is advisable to concentrate the manuscript on a specific type of cancer rather than too many cell types, where the proposed mechanism may not be conserved. This focused approach will enhance the ability to unravel the function of the YY1-KDM5C complex within a more defined context and will contribute to drawing more solid conclusions. Thus, I recommend that the authors narrow their study to one type of cancer (renal? cervical?) and use the normal counterpart as previously suggested (point 1)

Response: Thanks for the valuable suggestion. We have narrowed our study to renal cancer cells in the revised manuscript, and we have used a normal counterpart as requested by the reviewer, which is a human kidney proximal tubular epithelial cell line HK-2.

3. In Figure 3C, among the 4928 regions where YY1 binding decreases in the absence of KDM5C,

i) what about YY1 levels in regions that do not overlap with KDM5C (9473 regions)? Are these regions not affected by the knockdown?

Response: We analyzed the YY1 levels in regions that do not overlap with KDM5C, and it appears that YY1 chromatin binding exhibits similar pattern as in YY1 and KDM5C co-occupied regions (Fig R2). We speculated that KDM5C mediated YY1 enrichment in co-occupied region might serve as an initial step of YY1 chromatin binding, which is followed by other actions, such as CTCF-dependent looping^[7] and YY1 binding to specific DNA motifs^[8].

Fig R2. YY1 chromatin binding peaks in regions that do not overlap with KDM5C. Heatmap and metaplot showed YY1 chromatin enrichment at YY1 and KDM5C non-overlapping regions (n=9473) in ACHN cells with or without KDM5C.

ii) Considering the authors' proposition that YY1 binding depends on KDM5C and given YY1's ability to bind DNA to a specific DNA sequence, are the regions where both KDM5C and YY1 overlap enriched in YY1 DNA binding motifs? Alternatively, could they be enriched in another DNA binding motif or transcription factor (TF) that could serve as a connecting element between KDM5C and YY1? How would this crosstalk occur mechanistically?

Response: Thanks for the constructive comments. We further analyzed the ChIP-seq results by HOMER tool to delineate the enrichment of YY1 in different DNA motifs. The results showed that 1.7% regions where both KDM5C and YY1 overlap containing classic YY1 binding motif (Fig R3). KDM5C expression increased this ratio and loss of KDM5C reduced it. This result may suggest that the YY1 chromatin binding through interaction with KDM5C is an initial step for YY1 to exert the transcription function, which is followed by binding to different DNA motifs.

Fig R3. The proportion of the most classical YY1 binding motifs in KDM5C and YY1 overlapping regions

YY1 chromatin enrichment with classical binding motifs at YY1 and KDM5C overlapping regions in ACHN cells with or without KDM5C.

iii) The authors should link ChIP-seq (Figure 4) with RNA-seq results (Figure 5). Are genes affected by knockdowns significantly enriched in YY1 and KDM5C at the promoters?

Response: We appreciate your comments. We linked ChIP-seq with RNA-seq results and confirmed that these genes affected by knockdowns are enriched in YY1 and KDM5C at their promoters (Fig EV5).

4. The KDM5C ChIP-seq data are obtained from GSE71327, primarily performed on ZR-75-30, an epithelial cell line isolated from the mammary gland of a ductal carcinoma patient. On which cells have the experiments included in the manuscript been performed? Is there any reason not to utilize data from a system that is more compatible with the study in question?

Response: We thank the constructive suggestion. We tried to utilize data from kidney tissue, but such kind of data is not available. In our study, we obtained KDM5C ChIP seq data from a published database GSE71327, which has been extensively cited by numerous studies^[9]. We have also attempted to generate high quality ChIP-seq data in renal cancer cells and several KDM5C antibodies (Abcam, ab194288; Bethyl, A301-034A; Proteintech, 14426-1-AP) were tested, unfortunately, none of them was able to pull-down enough genomic DNA in renal cancer cells.

5. In Figure 4A, the authors conclude that "ectopic KDM5C enhances YY1 chromatin binding," but the data presented show a very weak effect (despite the strong overexpression of KDM5C, Fig S2A). Could the authors clarify this point?

Response: Thanks for your comments. Statistical analysis of the difference of YY1 chromatin binding was performed. The result was statistically significant ($P < 0.0001$). It confirmed that ectopic KDM5C enhances YY1 chromatin binding (Fig R4).

Fig R4. The distribution of YY1 peaks on YY1 and KDM5C overlapping regions (n=4928) in ACHN cells expressing KDM5C-WT

6. In Figure 5, the cell model used for RNA-seq analysis is not mentioned. This information is vital as the authors aim to connect the expression results with the functional data presented in Figure 2.

Response: We apologize for the negligence. Our RNA-seq were performed in renal cancer cell ACHN, which has been added in the legend. According to the reviewers' suggestion, we performed additional experiments with ACHN cancer cells, including colony formation, cell proliferation, tumor xenograft and et al, which were presented in the revised manuscript (Fig 2B-D, 4G and 4J).

Minor points:

1-In Figure 2A, the levels of YY1 should be displayed.

Response: We rearranged Fig 2A and provided the protein level of KDM5C and YY1 in human kidney proximal tubular epithelial cell HK-2 and renal cancer cell lines.

2-Figure 6 should be moved to Supplementary.

Response: Thanks for your suggestion. Fig 6 was moved to Supplementary Fig EV5 as requested.

3-In Figure 7C, the model is unclear, and the authors should revise it to provide a more straightforward and comprehensible representation that succinctly summarizes the conclusions of their work.

Response: We thank the constructive suggestion and modified our proposed model in a more straightforward and comprehensible way.

Referee #3:

In this work, the authors presented data showing a synthetic lethal interaction between YY1 and KDM5C and the inhibition of both could be a potential combination therapy for cancer treatment. They showed that YY1 knockdown only affected tumor cell proliferation when KDM5C expression was low or knocked down. Mechanistically, KDM5C can recruit YY1 to the promoters of several cell-fate essential genes and regulate their expression. The findings are interesting and novel. However, the overall writing needs to be improved and some points are not adequately supported. Moreover, the major concern I have is that the cell line used for mechanistic study is not one of the cell lines used for phenotypic characterization.

Major points:

1. In Fig1, the authors showed the interaction between KDM5C and YY1 in 293T cells exogenously and endogenously. Is this interaction also present in CAA3 and 769P cells? In Fig1F in the input blot, why is the background in each lane different? Some of the lanes even share some common bands. To address this, a control with an empty vector needs to be included.

Response: According to the reviewers' suggestion, we performed co-IP experiment in renal carcinoma cell lines 769-P, as shown in Fig 1E in the revised manuscript. KDM5C bound to YY1 in 769-P cells. We also repeated co-IP experiment in Fig.1F, the empty vector control was added according to reviewer's suggestion, which was showed in Fig 1G in the revised manuscript.

2. In Fig3, the authors used ChIP-seq to explore the underlying mechanism. Why did the authors choose to use ACHN cell line? The authors need to show the phenotype in ACHN caused by KDM5C and YY1 depletion or conduct

ChIP-seq in CAA3 and 769P cells. Similarly in Fig3B, what cell line was used in KDM5C ChIP-seq? The cell lines used need to be consistent throughout the study.

Response: We appreciated your constructive comments. Because VHL is a typical mutated gene in renal cancer, and ACHN is a VHL wide type cell line. To avoid any possible disturbance that may be caused by VHL depletion, we chose ACHN cell for ChIP-seq assay. According to the reviewers' suggestion, we have performed additional experiments with ACHN renal cancer cells, including colony formation, cell proliferation and tumor xenograft, which were shown in Fig 2B-D, 4G and 4J in the revised manuscript.

In Fig3B, we obtained KDM5C ChIP-seq data from a published database GSE71327, which is derived from a breast cancer cell line and has been extensively cited by numerous studies^[9]. We have also attempted to generate high quality ChIP-seq data in renal cancer cells and several KDM5C antibodies (Abcam, ab194288; Bethyl, A301-034A; Proteintech, 14426-1-AP) were tested, unfortunately, none of them was able to pull-down enough genomic DNA in renal cancer cells.

3. In Fig4, the authors focused a lot on the H514A mutant of KDM5C. The authors showed that H514A failed to increase YY1 recruitment like WT KDM5C and H3K4me3 does not affect YY1 recruitment. However, H514A can still interact with YY1 and it can partially rescue the phenotype in RRC4 but not in CaSki, which is very confusing. Could the authors put KDM5C-WT or KDM5C-H514A back to KDM5C and YY1 double knockdown CAA3 and 769P cells to clarify this? To explain this, the authors claimed that H514A "influenced the chromatin binding activity of KDM5C". This claim was only backed up by Fig4F, which is not sufficient, and it goes against a published study(PMID: 33977073). Much work needs to be done to support this claim.

Response: We agreed with your suggestion. We put KDM5C-WT or KDM5C-H514A back to KDM5C and YY1 double knockdown ACHN and 769-P tumor cells to further confirm the role of KDM5C-H514A in recruiting YY (Fig 4G-H and S1B-C)¹. Tumor cell proliferation was markedly inhibited when both of KDM5C and YY1 were depleted. The inhibitory effect was blunted when KDM5C-WT is re-expressed; in contrast, KDM5C-H514A only moderately restored the growth of ACHN and 769-P double knockdown cells, suggesting that KDM5C-H514A is defective in recruiting YY1. Our result showed that the protein levels of KDM5C WT and H514A in whole-cell lysates were indistinguishable, but H514A exhibited obviously reduced chromatin-binding capacity (Fig 4F). We speculated that H514A might destroy the pocket structure that accommodating H3K4me_{2/3}.

In addition, we read the published study (PMID: 33977073)^[10], which focused on KDM5C mediated estrogen receptor enhancer recruitment. Because

enhancer elements contain much less H3K4me2/3 compared with promoters, it is likely that KDM5C bind to enhancers via other mechanism, such as by interacting with RACK7^[9].

4. Again, the cell lines used for RNA-seq analysis is not specified. What pathways are enriched upon KDM5C and YY1 double knockdown? Also, there is an inconsistency between the qPCR and RNA-seq data. In the RNA-seq result, SMC2 and SMC4 is the highest in shYY1 group and LMNB is the lowest in shKDM5C group. However, the qPCR results showed a different trend. Are these genes also downregulated in double knockout CAA3 and ACHN cells?

Response: Thanks for the suggestion. The RNA-seq analysis was conducted in ACHN cell line, which is a VHL proficient renal cancer cell line and has been employed in colony formation, cell proliferation and tumor xenograft experiments in the revised manuscript (Fig 2B-D, 4G and 4J).

KEGG analysis showed the enrichment pathway was closely associated with cell cycle and DNA replication, suggesting that the synergy of these two genes may play an important role in regulating cell cycle and apoptosis (Fig R5)

To address the reviewers' concern, we applied qPCR in ACHN cells (Fig 5E) and compared the results with RNA-seq data. The qPCR results appear to be consistent with the observations in RNA-seq analysis. Some inconsistency shown in previous work may be due to the different cell lines (769-P and ACHN) used in those experiments.

Fig R5. KEGG pathway enrichment analysis of YY1 and KDM5C double knockdown ACHN cells

5. In Fig6A-B and FigS3, the KDM5C-H514A group seems to be left out. Did

KDM5C H514A change the size of YY1 peaks? Also the authors need to verify these results using ChIP-qPCR. To add to the robustness of the data, the authors could also perform ChIP-qPCR in different cell lines.

Response: We added the KDM5C-H514A group and presented it in Fig EV5A-B of revised manuscript. Consistent with the ChIP-seq result, KDM5C-H514A marginally increase the YY1 intensity at indicated regions.

In response to other reviewer's suggestion, we performed the ChIP-seq experiments targeting YY1 in HK-2 (human kidney proximal tubular epithelial cell line) in the presence or absence of KDM5C, which is presented in Fig EV2A. Moreover, ChIP-qPCR was performed and the result showed the YY1 enrichment at promoter region of indicated genes was increased after KDM5C-WT was expressed, whereas KDM5C-H514A failed to increase YY1 binding (Fig EV2C).

Minor points:

1. The figure legend of 2D and 2E was switched.

Response: Thanks for pointing out the mistakes, which has been corrected in the revised manuscript. We will double check the manuscript to avoid similar errors.

2. The western blot images could be cropped taller generally. Some bands are cropped out.

Response: Thanks for your suggestion. The western blot images have been rearranged.

3. In page 7, it says "the intensity of YY1 binding peaks in both KDM5C-depleted and parental ACHN was reduced upon loss of KDM5C". Do the authors mean "the intensity of YY1 binding peaks was reduced upon loss of KDM5C"?

Response: We corrected this incorrect description.

4. 769-P cell line is sometimes labeled 769P.

Response: Thanks for your comments. We corrected those mistakes.

5. More details are generally needed in the method section. Some coIP experiments and ChIP experiments weren't included. ACHN KDM5C-H514A was left out in the ChIP method.

Response: Thanks for your suggestion. We modified the corresponding description in the methods section.

Once again, thank you very much for your comments and suggestions.

Looking forward to your response!

With kind personal regards.

Yours sincerely,

Jie Xiong

References:

- [1]. Yaping Zhang, et al. JAM3 maintains leukemia-initiating cell self-renewal through LRP5/AKT/ β -catenin/CCND1 signaling. *J Clin Invest.* 2018;128(5):1737-1751.
- [2]. Hui Tong, et al. Silencing of KNTC1 inhibits hepatocellular carcinoma cells progression via suppressing PI3K/Akt pathway. *Cell Signal.* 2023; 101:110498.
- [3]. Verónica Dávalos, et al. Human SMC2 protein, a core subunit of human condensin complex, is a novel transcriptional target of the WNT signaling pathway and a new therapeutic target. *J Biol Chem.* 2012;287(52):43472-81.
- [4]. L Jiang, et al. Overexpression of SMC4 activates TGF β /Smad signaling and promotes aggressive phenotype in glioma cells. *Oncogenesis.* 2017;6(3):e301.
- [5]. Yongyu Yang, et al. Lamin B1 is a potential therapeutic target and prognostic biomarker for hepatocellular carcinoma. *Bioengineered.* 2022;13(4):9211-9231.
- [6]. Andrei L Gartel. FOXM1 in Cancer: Interactions and Vulnerabilities. *Cancer Res.* 2017;77(12):3135-3139.
- [7]. Jonathan A Beagan, et al. YY1 and CTCF orchestrate a 3D chromatin looping switch during early neural lineage commitment. *Genome Res.* 2017;27(7):1139-1152.
- [8]. Jeong do Kim, et al. YY1's longer DNA-binding motifs. *Genomics.* 2009;93(2):152-8.
- [9]. Hongjie Shen, et al. Suppression of Enhancer Overactivation by a RACK7-Histone Demethylase Complex. *Cell.* 2016;165(2):331-42.
- [10]. Hai-Feng Shen, et al. The Dual Function of KDM5C in Both Gene Transcriptional Activation and Repression Promotes Breast Cancer Cell Growth and Tumorigenesis. *Adv Sci (Weinh).* 2021;8(9):2004635.

Dear Mrs. Xiong,

Thank you for the submission of your revised manuscript to EMBO reports. We have now received the full set of referee reports that is pasted below.

As you will see, while referee 1 is happy and referee 3 has more minor last comments, referee 2 is asking for further experimentation. I discussed this set of comments with my colleagues here, and we decided that we can overrule referee 2's comments on the synthetic lethality in non-tumor cells, which is not a concern from our side.

However, we do agree with referee 2 that "KDM5C ChIP-seq is a key result that should be performed in the same system in which the YY1 ChIP-seq was done". Given that the abstract of your ms mentions as one main finding that KDM5C is essential for global YY1 chromatin recruitment, showing that both factors co-localize on chromatin in the same system is indeed an important aspect of your paper.

I would like to know whether you will be able to perform and add these experiments, and also compare the ChIP-seq data with your RNA-seq results. Please let me know what you think. We can discuss this also further in a video chat, if you like.

Please also note that in all (relevant) figure legends exact p-values need to be provided, statistical tests, error bars and "n" need to be specified, and box plots need to be defined in terms of minima, maxima, centre, bounds of box and whiskers and percentile. This information is currently incomplete.

Assuming that we can agree on a final set of experimental revisions, I look forward to seeing a newly revised form of your manuscript when it is ready.

Referee #1:

The revised manuscript addressed all of previous critiques and should be publishable.

Referee #2:

The authors have revised their article and conducted some of the experiments suggested by this reviewer. The new results indicate that synthetic lethality between YY1 and KDM5C also operates in normal human renal HK-2 cells, suggesting that this mechanism is not necessarily exclusive to or specific for tumor cells. In fact, normal cells proliferate significantly less with the double knockdown (Figure R1). These results, at the very least, limit the potential use of combination treatments that target this therapeutic vulnerability and require the authors to reconsider their conclusions.

Regarding the mechanism involved, the observation that YY1 depletion occurs upon shKDM5C knockdown even in regions without overlapping between YY1 and KDM5C (Figure R2) points to an indirect mechanism. However, these conclusions are not sufficiently robust because the KDM5C binding profile was derived from public data obtained from a breast cancer cell line (GSE71327), which is markedly different from the renal cell model the authors aim to investigate. In this reviewer's opinion, KDM5C ChIP-seq is a key result that should be performed in the same system where the YY1 ChIP-seq was done. Although this experiment may be challenging, the authors could attempt to use tagged versions of KDM5C that facilitates immunoprecipitation. Since it's a part of synthetic lethality, the authors must produce original data on KDM5C's genome-wide binding in renal tumor cells.

The connection between the ChIP-seq and RNA-seq results requested in point 3.iii should be addressed globally, with the appropriate controls, rather than in isolated genes as shown in the new Figure EV5. It should be evaluated whether YY1 and KDM5C are enriched at the promoters of all genes affected by the respective knockdowns, and determine if this enrichment varies in unaffected genes.

Referee #3:

In this revision, the authors addressed most of the raised points but not all.

In the added rescue experiments in Fig G-H and J, KDM5C was re-expressed in KDM5C and YY1 double knockdown cells. However, there is no mention in the method section how these constructs are made to be shRNA resistant. In addition, what antibiotic resistance does pHAGE-KDM5C have? Only puromycin was used to select cells with knockdown and overexpression?

In Fig EV5B, when comparing KDM5C and H514, H514 increased YY1 enrichments on these gene promoters to a greater extent than KDM5C, which is against the authors' claim that an intact JmjC domain is required for KDM5C-mediated YY1 recruitment for these genes.

In the added CHIP-qPCR experiments in Fig EV2B-C, all of which were performed in HK-2 cells, the results in ACHN weren't verified. The figure legends are not correct in that no double-knockdown cells were used here. IgG control should also be included here as negative controls.

The language especially in the newly added method section needs work. Most sentences are grammatically incorrect.

Dear Dr Schnapp,

We greatly appreciate all the constructive comments on our manuscript (EMBOR-2024-58727V2). We have performed additional experiments and prepared a revised manuscript according to the suggestion. **Below is our response (in blue) to reviewers' comments (in black).** Please feel free to contact me if you have further questions or need additional information.

Referee #1:

The revised manuscript addressed all of previous critiques and should be publishable.

Referee #2:

The authors have revised their article and conducted some of the experiments suggested by this reviewer. The new results indicate that synthetic lethality between YY1 and KDM5C also operates in normal human renal HK-2 cells, suggesting that this mechanism is not necessarily exclusive to or specific for tumor cells. In fact, normal cells proliferate significantly less with the double knockdown (Figure R1). These results, at the very least, limit the potential use of combination treatments that target this therapeutic vulnerability and require the authors to reconsider their conclusions.

Regarding the mechanism involved, the observation that YY1 depletion occurs upon shKDM5C knockdown even in regions without overlapping between YY1 and KDM5C (Figure R2) points to an indirect mechanism. However, these conclusions are not sufficiently robust because the KDM5C binding profile was derived from public data obtained from a breast cancer cell line (GSE71327), which is markedly different from the renal cell model the authors aim to investigate. In this reviewer's opinion, KDM5C ChIP-seq is a key result that should be performed in the same system where the YY1 ChIP-seq was done. Although this experiment may be challenging, the authors could attempt to use tagged versions of KDM5C that facilitates immunoprecipitation. Since it's a part of synthetic lethality, the authors must produce original data on KDM5C's genome-wide binding in renal tumor cells.

Response: Thanks for the constructive suggestion. In order to investigate the genomic binding of KDM5C in renal cancer cells, we performed the Cleavage Under Targets and Tagmentation (CUT&Tag) assay in ACHN renal cell line. ChIP-grade anti-KDM5C antibody (Bethyl, A301-034A) and anti-YY1 (CST, D5D9Z) were used for specific immunoprecipitation. Rabbit IgG (ABclonal, AC042) was used as a negative control. We interrogated the CUT&Tag data, and the annotation of the YY1 and KDM5C peaks revealed a substantial overlap

(n=3645) between YY1 and KDM5C binding events (Figure 3B) , which is consistent with our previous data. Similarly, the heatmap and metaplot analysis showed that the enrichment of YY1 was reduced after KDM5C was depleted in the peaks where YY1 and KDM5C directly related (Figure 3C). Our results demonstrated that YY1 chromatin binding was partially relying on the KDM5C protein level.

The connection between the ChIP-seq and RNA-seq results requested in point 3.iii should be addressed globally, with the appropriate controls, rather than in isolated genes as shown in the new Figure EV5. It should be evaluated whether YY1 and KDM5C are enriched at the promoters of all genes affected by the respective knockdowns, and determine if this enrichment varies in unaffected genes.

Response: According to the suggestion, we compared ChIP-seq and RNA-seq data, and found that almost half of the downregulated genes (49.96%) in KDM5C/YY1 double knockdown group were annotated in the genomic regions where YY1 and KDM5C overlapped, as shown in Fig EV5B in the revised manuscript, reinforcing that ideal that KDM5C globally increases the expression of essential cell-fate genes by recruiting YY1 to their promoters.

Referee #3:

In this revision, the authors addressed most of the raised points but not all.

In the added rescue experiments in Fig G-H and J, KDM5C was re-expressed in KDM5C and YY1 double knockdown cells. However, there is no mention in the method section how these constructs are made to be shRNA resistant. In addition, what antibiotic resistance does pHAGE-KDM5C have? Only puromycin was used to select cells with knockdown and overexpression?

Response: Thanks for the reviewer's suggestion. We used a GFP labeled KDM5C knockdown plasmid to generate the double knockdown cells. For rescue experiment, a hygromycin resistance gene was inserted into pHAGE-KDM5C/H514A-puro plasmid for screening. We added the corresponding description in the methods section of the revised manuscript.

In Fig EV5B, when comparing KDM5C and H514, H514 increased YY1 enrichments on these gene promoters to a greater extent than KDM5C, which is against the authors' claim that an intact JmjC domain is required for KDM5C-mediated YY1 recruitment for these genes.

Response: Thanks for the reviewer's suggestion. Our results indicated that KDM5C WT rather than H514A promoted the enrichment of YY1 chromatin binding. The YY1 peaks of H514A group might be slightly shift when comparing

with control or WT group in some of the genes (eg, LMNB1, KNTC1, SMC2 and SMC4, Fig R1). We speculated that mutation in H514 might alter the KDM5C chromatin binding pattern and the associated YY1 recruitment at promoters, indicating that JmjC domain is important for KDM5C-mediated YY1 recruitment.

Fig R1 The effect of KDM5C on YY1 enrichment at promoter regions of specific genes

In the added ChIP-qPCR experiments in Fig EV2B-C, all of which were performed in HK-2 cells, the results in ACHN weren't verified. The figure legends are not correct in that no double-knockdown cells were used here. IgG control should also be included here as negative controls.

Response: Thanks for the reviewer's suggestion. The genomic binding of KDM5C and YY1 were evaluated by using Cleavage Under Targets and Tagmentation (CUT&Tag) assay in ACHN renal cell line in the revised manuscript (Figure 2B).

We have corrected the error in Fig EV2. According to the suggestion, we re-calculated the fold enrichment of those genes in YY1-ChIP, and IgG was used as a negative control. The calculation formula is as followed:

$$\Delta Ct [\text{normalized ChIP}] = (Ct [\text{ChIP}] - (Ct [\text{Input}] - \text{Log}_2 (\text{Input Dilution Factor})))$$

$$\Delta\Delta Ct [\text{ChIP/NIS}] = \Delta Ct [\text{normalized ChIP}] - \Delta Ct [\text{IgG}]$$

$$\text{Fold Enrichment} = 2^{(-\Delta\Delta Ct [\text{ChIP/NIS}])}$$

The figures were rearranged in the revised manuscript (Figure EV2B-C).

The language especially in the newly added method section needs work. Most sentences are grammatically incorrect.

Response: Thanks for the comments. We carefully read the entire text and corrected those grammar errors.

Once again, thank you very much and looking forward to your response!

With kind personal regards.

Yours sincerely,

Jie Xiong

Dear Ms. Yang,

Thank you for the submission of your newly revised manuscript. Both referees 2 and 3 have seen it now, and while referee 3 is satisfied with the revisions, referee 2 is asking for some more data, please see below.

I would like to invite you to address these last comments and send us a newly revised manuscript as soon as possible, together with a point-by-point response to all last comments.

A few editorial requests also remain to be addressed:

- The manuscript sections should be in the following order: Title page - Abstract & Keywords - Introduction - Results - Discussion - Methods - Data Availability - Acknowledgments - Disclosure Statement & Competing Interests - References - Figure Legends - (Main Tables with legends) - Expanded View Figure Legends.
- The Competing interest declaration needs to be renamed to "Disclosure Statement and Competing Interests"
- The Author Contributions need to be removed from the ms file. All contributions are entered online during ms submission.
- Please add direct URLs for the GEO datasets to the DAS (Data Availability Section).
- These callouts need to be corrected since there aren't any "S" figures: "S1B-D" and "S1E-F".
- The Supplemental data need to be renamed to "Expanded View Figure Legends" in the ms file.
- Source data (SD): SD for the main figures need to be uploaded as 1 folder per figure, while the SD for EV figures can be grouped together in 1 file.
- The Reagents and Tools table needs to be provided as a separate file.
- "URL links should resolve to a dataset-specific page (not to the generic database homepage)." ==> Please note that a pubmed article URL is provided for (Data ref: Shen et al, 2016) data citation, please correct.
- Please add the exact p values to the legends of figures 2c-d; 3a; 4g-i; 5a-d; EV 4a-c.
- Please indicate the statistical test used for data analysis in the legends of figures 1a; 5a-d; EV 4a-c.
- Please note that the box plots need to be defined in terms of minima, maxima, centre, bounds of box and whiskers, and percentile in the legend of figure 3a.
- Please note that information related to n is missing in the legend of figure 1a.
- Please note that the error bars are not defined in the legends of figures 4g-i; 5f; 6a-b; EV 2b-c.
- Please note that the measure of center for the error bars needs to be defined in the legend of figure 1a.
- It would be helpful and beneficial if a native speaker could proof-read your full manuscript.

I would like to suggest some changes to the abstract that needs to be written in present tense. Please let me know whether you agree with the following:

An understanding of the enzymatic and scaffolding functions of epigenetic modifiers is important for the development of epigenetic therapies for cancer. The H3K4me2/3 histone demethylase KDM5C has been shown to regulate transcription. The diverse roles of KDM5C are likely determined by its interacting partners, which are still largely unknown. In this study, we screen for KDM5C-binding proteins and show that YY1 interacts with KDM5C. A synergistic antitumor effect is exerted when both KDM5C and YY1 are depleted, and targeting YY1 appears to be a vulnerability in KDM5C-deficient cancer cells. Mechanistically, KDM5C promotes global YY1 chromatin recruitment, especially at promoters. Moreover, an intact KDM5C JmjC domain but not KDM5C histone demethylase activity is required for KDM5C-mediated YY1 chromatin binding. Transcriptional profiling reveals that dual inhibition of KDM5C and YY1 increases transcriptional repression of cell cycle- and apoptosis-related genes. In summary, our work demonstrates a synthetic lethal interaction between YY1 and KDM5C and suggests combination therapies for cancer treatments.

EMBO press papers are accompanied online by A) a short (1-2 sentences) summary of the findings and their significance, B) 2-3 bullet points highlighting key results and C) a synopsis image that is exactly 550 pixels wide and 200-600 pixels high (the height is variable). The synopsis image should provide a sketch of the major findings, like a graphical abstract. Please note that text needs to be readable at the final size. Please send us this information along with the final manuscript.

I look forward to seeing a newly revised version of your manuscript as soon as possible.

Referee #2:

In response to this reviewer, the authors have addressed the issue by monitoring the binding of YY1 through the CUT&Tag assay in renal tumor cells, as presented in the new Figure 3B. The text accompanying the figure states: "We performed the Cleavage Under Targets and Tagmentation (CUT&Tag) assay in the ACHN renal cell line, and the annotation of the YY1 and KDM5C peaks revealed substantial overlap (n=3645) between YY1 and KDM5C binding events (Figure 3B)." However, the authors must provide a statistical assessment of the significance of this overlap, considering that the 3645 overlapping regions account for only a portion of the 24,100 total binding regions for KDM5C. Including the expected random overlap value would help justify the use of the term "substantial."

Additionally, to ensure that the CUT&Tag experiments were successful, especially given that the protocol does not involve protein-DNA crosslinking, the authors must present a list of enriched DNA binding motifs from these experiments and confirm whether they align with what is expected for a chromatin precipitation involving KDM5C. Furthermore, the number of replicates for the CUT&Tag experiments is not mentioned and should be clarified.

Regarding the concerns raised by reviewer 3, the authors have adequately addressed the methodological points. However, in my opinion, the issue concerning Figure EV5B remains unresolved. While the YY1 peak in KDM5C H514A appears broader compared to the Parental line, there is no clear decrease in YY1 levels. Therefore, it would be appropriate that reviewer #3 provide the final assessment on this matter.

Referee #3:

The authors have addressed my raised points.

As for the role of the Jmjc domain, in Fig EV5A it was shown H514A did not increase YY1 binding compared with Wild type KDM5C. While for some individual genes shown in Fig C-D, only JAM3 supports the author's conclusion, which is not ideal. However, the rescue experiments in Fig 4G-J and the ChIP-qPCR in Fig EV2C provided more evidence supporting this conclusion.

The writing should be improved before publication.

Dear Dr. Schnapp,

We greatly appreciate the reviewers' constructive suggestion and comments on our manuscript (EMBOR-2024-58727V3). **We have prepared a revised manuscript according to your suggestion and reviewers' comments. Below is our response (in blue) to reviewers' comments (in black).** Please feel free to contact me if you have further questions or need additional information.

Referee #2:

In response to this reviewer, the authors have addressed the issue by monitoring the binding of YY1 through the CUT&Tag assay in renal tumor cells, as presented in the new Figure 3B. The text accompanying the figure states: "We performed the Cleavage Under Targets and Tagmentation (CUT&Tag) assay in the ACHN renal cell line, and the annotation of the YY1 and KDM5C peaks revealed substantial overlap (n=3645) between YY1 and KDM5C binding events (Figure 3B)." However, the authors must provide a statistical assessment of the significance of this overlap, considering that the 3645 overlapping regions account for only a portion of the 24,100 total binding regions for KDM5C. Including the expected random overlap value would help justify the use of the term "substantial."

Response: Thanks for the constructive suggestion. We performed the CUT&Tag assay in ACHN cells, revealing that there are 3645 overlapped peaks between YY1 and KDM5C binding events. In order to address the reviewer's concern, we further analyzed the overlapping genes within these binding events, as illustrated in Figure R1. Remarkably, we found that 7957 genes were concurrently bound by both YY1 and KDM5C (Hypergeometric test, $P=4.87e-17$), which account for 56.32% of the total KDM5C bound genes and 66.71% of the total YY1 bound genes. Moreover, we calculated the correlation of KDM5C and YY1 genome-wide binding intensity by using Pearson Correlation. As shown in Figure R2, our data showed a strong positive correlation of KDM5C and YY1 genome-wide binding signal, again highlighting the coordinated actions of these two regulators at the genomic level. To avoid unnecessary confusion, we replaced "substantial" with "obvious".

Fig R1 Venn diagram showing the genes number in YY1 and KDM5C co-occupied regions.

Fig R2 The correlation of genome-wide KDM5C and YY1 binding intensity was calculated by Pearson Correlation.

Additionally, to ensure that the CUT&Tag experiments were successful, especially given that the protocol does not involve protein-DNA crosslinking, the authors must present a list of enriched DNA binding motifs from these experiments and confirm whether they align with what is expected for a chromatin precipitation involving KDM5C. Furthermore, the number of replicates for the CUT&Tag experiments is not mentioned and should be clarified.

Response: According to the suggestion, we firstly generated a list containing enriched DNA binding motifs derived from our CUT&Tag experiment results (Excel file will be attached to our email to editor). Those motifs were then analyzed in the ChIP-seq data of KDM5C (Data ref: Shen *et al*, 2016). As shown in Fig R3, the KDM5C bound motifs derived from CUT&Tag or ChIP-seq data exhibit quite similar pattern. Besides, our CUT&Tag data indicated that KDM5C primarily binds to promoters and enhancers, which is consistent with previous study (Outchkourov *et al*, 2013; Shen *et al*, 2016), indicating that our CUT&Tag data is reliable. When performing CUT&Tag experiments, triplicate samples of each group were subjected to quality control test, and subsequently, two samples from each group were randomly chosen for DNA sequencing (we added n=2 in CUT&Tag experiments in the revised manuscript).

Fig R3 CUT&Tag results align with the ChIP-seq data of KDM5C (GSE71327)
 A Genome-wide annotation of CUT&Tag or ChIP-seq data of KDM5C (GSE71327)
 B Venn diagram showing the overlapping motifs number of our own CUT&Tag results and the ChIP-seq data of KDM5C (GSE71327)

Regarding the concerns raised by reviewer 3, the authors have adequately addressed the methodological points. However, in my opinion, the issue concerning Figure EV5B remains unresolved. While the YY1 peak in KDM5C H514A appears broader compared to the Parental line, there is no clear decrease in YY1 levels. Therefore, it would be appropriate that reviewer #3 provide the final assessment on this matter.

Response: Thanks for the comments. As mentioned by reviewer #3, in addition to the ChIP-seq data presented in Figure EV5B, other experiments, including proliferation assays in Figure 4G-J and ChIP-qPCR analysis in Figure EV2C, were sufficient to support our conclusion.

Referee #3:

The authors have addressed my raised points.

As for the role of the Jmjc domain, in Fig EV5A it was shown H514A did not increase YY1 binding compared with Wild type KDM5C. While for some individual genes shown in FigC-D, only JAM3 supports the author's conclusion, which is not ideal. However, the rescue experiments in Fig 4G-J and the ChIP-qPCR in Fig EV2C provided more evidence supporting this conclusion.

The writing should be improved before publication.

Response: We appreciate the reviewer's comments. We have invited a native English speaker to help us to improve the writing.

References

- Data ref: Shen H, Xu W, Guo R, Rong B, Gu L, Wang Z, He C, Zheng L, Hu X, Hu Z *et al*
Suppression of Enhancer Overactivation by a RACK7-Histone Demethylase Complex (2016)
Gene Expression Omnibus GSE71327 (<https://www.ncbi.nlm.nih.gov/pubmed/27058665>)
[DATASET]
- Outchkourov NS, Muino JM, Kaufmann K, van Ijcken WF, Groot Koerkamp MJ, van Leenen D,
de Graaf P, Holstege FC, Grosveld FG, Timmers HT (2013) Balancing of histone H3K4
methylation states by the Kdm5c/SMCX histone demethylase modulates promoter and
enhancer function. *Cell Rep* 3: 1071-1079
- Shen H, Xu W, Guo R, Rong B, Gu L, Wang Z, He C, Zheng L, Hu X, Hu Z *et al* (2016)
Suppression of Enhancer Overactivation by a RACK7-Histone Demethylase Complex. *Cell*
165: 331-342

Dear Dr. Xiong,

Thank you for the submission of your newly revised manuscript. We have now received the comments from referee 2 who is satisfied with the final revisions. Please do address though her/his last comments listed below.

I also slightly changed your short summary and bullet points:

This study shows that KDM5C binds to YY1, that it facilitates YY1 chromatin recruitment, and that targeting YY1 increases the vulnerability of KDM5C-low cancer cells.

- KDM5C promotes global YY1 chromatin recruitment, especially at promoters.
- An intact JmjC domain is required for KDM5C mediated YY1 chromatin binding.
- Dual inhibition of KDM5C and YY1 results in transcriptional repression of cell cycle-and apoptosis-related genes and has a synergistic antitumor effect

The synopsis image you sent is fine.

Referee #2:

The authors have adequately addressed my previous concerns. I recommend incorporating a paragraph similar to the one below to clarify the point regarding the overlap between YY1 and KDM5C:

The annotation of the YY1 and KDM5C peaks revealed a significant overlap, with 7,957 genes being concurrently bound by both YY1 and KDM5C (Hypergeometric test, $P=4.87e-17$). This overlap accounts for 56.32% of the total genes bound by KDM5C and 66.71% of the total genes bound by YY1.

Dear Dr. Schnapp,

We greatly appreciate the reviewers' constructive suggestion on our manuscript (EMBOR-2024-58727V4). **We have prepared a revised manuscript according to the reviewers' comments. Below is our response (in blue) to reviewers' comments (in black).** Please feel free to contact me if you have further questions or need additional information.

Referee #2:

The authors have adequately addressed my previous concerns. I recommend incorporating a paragraph similar to the one below to clarify the point regarding the overlap between YY1 and KDM5C:

The annotation of the YY1 and KDM5C peaks revealed a significant overlap, with 7,957 genes being concurrently bound by both YY1 and KDM5C (Hypergeometric test, $P=4.87e-17$). This overlap accounts for 56.32% of the total genes bound by KDM5C and 66.71% of the total genes bound by YY1.

Response: Thanks for the constructive suggestion. We added the description about overlap genes bound by both YY1 and KDM5C in our revised manuscript.

Dr. Jie Xiong
Wuhan university
China

Dear Dr. Xiong,

I am very pleased to accept your manuscript for publication in the next available issue of EMBO reports. Thank you for your contribution to our journal.
